Dual-Wavelength Radar Technique Development for Snow Rate Estimation: A Case Study from GCPEx
Gwo-Jong Huang
Viswanathan N. Bringi
Andrew J. Newman
GyuWon Lee
Dmitri Moisseev
Branislav M. Notaroš





systems (ground-based, airborne or satellite) have allowed for observations of the complexity inherent in winter precipitation via dedicated field programs (e.g., Skolfronik-Jackson et al. 2015; Petaja et al. 2016). These large field programs are vital given that the retrieval problem is severely under-constrained due to large number of geometrical and microphysical parameters of natural snowfall, their extreme sensitivity to subtle changes in environmental conditions, and co-existence of

different populations of particle types within the sample volume (e.g., Syzrmer and Zawadzki 2014).

The surface imaging instruments that give complementary measurements and used in a number of recent studies include (i) 2D-Video Disdrometer (2DVD; Schöenhuber et al. (2008), (ii) Precipitation Imaging Package (PIP; von Lerber et al. 2017), (iii) Multi-Angle Snowflake Camera (MASC; Garrett et al. 2012). When these instruments are used in conjunction with a

well-shielded GEONOR or PLUVIO gauge, it is shown that a physically consistent representation of the geometric, microphysical, and scattering properties needed for radar-based QPE can be achieved (Szyrmer and Zawadzki 2010; Huang et al. 2015; von Lerber et al. 2017). In this study we use the 2DVD and PLUVIO gage located within a double fence international reference (DFIR) wind shield to reduce wind effects.

Radar-based QPE has generally been based on $Z_e$-$SR$ ($Z_e$: reflectivity; $SR$: liquid equivalent snow rate) power laws of the form $Z_e = \alpha(SR)^\beta$ where the pre-factor and exponent are estimated based on, (i) direct correlation of radar measured $Z_e$ with snow gauges (Rasmussen et al. 2003; Fujiyoshi et al. 1990; Wolfe and Snider 2012), or (ii) using imaging disdrometers such as 2DVD or PIP (Huang et al. 2015; von Lerber et al. 2017). Recently, Falconi et al. (2018) developed $Z_e$-$SR$ power laws at three frequencies (X, Ka and W-band) by direct correlation of radar and PIP observations. These studies have highlighted

the large variability of $\alpha$ due to particle size distribution (PSD), density, fall velocity, and dominant snow type, whereas the variability in $\beta$ is considerably less. Similarly, both methods, (i) and (ii), have been used to estimate ice water content ($IWC$) from $Z_e$ using power laws of the form $Z_e = a(IWC)^b$ based on airborne particle probe data, direct measurements of $IWC$, and airborne measurements of $Z_e$ (principally at X, $K_a$, and W-bands) (e.g., Heymsfield et al. 2005; Hogan et al. 2006; Heymsfield et al. 2016). The advantage of airborne data is that a wide variety of temperatures and cloud types can be

sampled (Heymsfield et al. 2017).

The dual-wavelength reflectivity ratio ($DWR$) radar-based QPE was proposed by Matrosov (1998; 2005) to improve $SR$ accuracy by estimating the PSD parameter (median volume diameter $D_0$) with relatively low dependence on density if assumed constant. There has been limited use of dual-λ techniques for snowfall estimation, mainly using vertical-pointing

ground radars or nadir pointing airborne radars (Liao et al. 2005, 2008, 2016; Szyrmer and Zawadzki 2014; Falconi et al. 2018). The dual-λ method is of interest to us due to the availability of the NASA D3R *scanning* radar (Vega et al. 2014), which, to the best of our knowledge, has not been exploited for snow QPE to date.



The *DWR* is defined as the ratio of the equivalent radar reflectivity factors at two different frequency bands. The main principle in *DWR* is that the particle's size-to-wavelength ratio falls in Rayleigh region at a low frequency band (e.g., Ku-band) but in Mie region at a high frequency band (e.g., Ka-band) (Matrosov 1998; Matrosov et al. 2005; Liao et al. 2016). Previous studies have shown that the *DWR* can be used to estimate $D_m$, where $D_m$ is defined as the ratio of 4th moment to 3rd moment of the PSD expressed in terms of liquid-equivalent size or mass (Liao et al. 2016). In this sense the *DWR* is similar to differential reflectivity ($Z_{dr}$) in dual-polarization radar technique where $Z_{dr}$ is used to estimate $D_m$ (but the physical principles are, of course, different; Meneghini and Liao 2007). The SR is obtained by 'adjusting' the coefficient $\alpha$ in the $Z_e$-SR power law based on the measure of $D_m$ provided by the *DWR*. The pre-factor $\alpha$ depends on the intercept parameter of PSD (von Lerber et al., 2017) and not on $D_m$ directly. However, because of apparent negative correlation between $D_m$ and PSD intercept parameter for a snowfall of a given intensity (Delanoe et al., 2005; Tiira et al., 2016) measurements of $D_m$ can be used to "adjust" $Z_e$-SR power law.

This paper is organized as follows. In Section 2, we introduce the approach and methodologies proposed and used in this study which may be considered as technique development. We briefly explain how to estimate mass of ice particles using a set of aerodynamic equations based on Böhm (1989) and Heymsfield and Westbrook (2010). We also give a brief introduction of the scattering model based on particle's mass. Section 3 provides a brief overview of instruments installed at the test site and the dual-wavelength radar used in this study (D3R: Vega et al. 2014). We analyze surface and D3R radar data from one synoptic snowfall event during GCPEx and compare *SR* retrieved from *DWR*-based relation with *SR* measured by a snow gauge. The conclusions and possibilities for further improvement of the proposed techniques are discussed in Section 4.

## 2 Methodology

### 2.1 Estimation of Particle Mass

The direct estimation of mass of an ice particle is difficult and at present there is no instrument available to do this automatically. The conventional method is to use a power law relation between mass and the maximum dimension of the particle of the form $m=aD^b$ where the pre-factor $a$ and exponent $b$ are computed via measurements of particle size distribution *N(D)* from aircraft probes and independent measurements of total ice water content as an integral constraint (Heymsfield et al. 2010). A similar method was used by Brandes et al. (2007) who used 2DVD data for *N(D)* and a snow gage for the liquid equivalent snow accumulation over periods of 5 min. These methods are more representative of an average relation when one particle type (e.g., snow aggregates) dominate the snowfall with large deviations possible for individual events with differing particle types (e.g., graupel).



To overcome these difficulties a more general method was proposed by Böhm (1989) based on estimating mass from fall velocity measurements, geometry and environmental data if the measured fall velocity is in fact the terminal velocity (i.e., in the absence of vertical air motion or turbulence and in more or less uniform precipitation). The methodology has been described in detail by (Szyrmer and Zawadzki 2010; Huang et al. 2015; von Lerber et al. 2017) and we refer to these articles

for details. The essential feature is the unique non-linear relation between the Davies (1945) number ($X$) and the Reynolds number ($Re$) where $X$ is the ratio of mass to area or $m/A_r^{0.25}$ ($A_r=A_e/A$ is the area ratio, where $A_e$ is the effective projected area normal to the flow and $A$ is the area of the minimum circumscribing circle or ellipse that completely contains $A_e$) and the $Re$ is the product of terminal fall speed and characteristic dimension of the particle. We have neglected the environmental parameters (air density, viscosity) as well as boundary layer depth of Abraham (1970) and the inviscid drag coefficient. The

procedure is to (i) compute $Re$ from fall velocity measurements and characteristic dimension of the particle (usually the maximum dimension), (ii) compute the Davies number $X$ which is expressed as a non-linear function of $Re$, and boundary layer parameters ($C_0 = 0.6$ and $\delta_0 = 5.83$; Böhm 1989), (iii) estimate particle mass from $X$ and $A_r$. Heymsfield and Westbrook (2010) proposed a simple adjustment (based on field and tank experiments) by defining a modified Davies number as proportional to $m/A_r^{0.5}$ along with different boundary layer constants ($C_0$=0.292; $\delta_0$=9.06) from Böhm. Their

adjustment was shown to be in very good agreement with recent tank experiments by Westbrook and Sephton (2017) especially for particles like pristine dendrites with low $A_r$ and at low $Re$. Note that the difference of $C_0$ and $\delta_0$ in Böhm and Heymsfield-Westbrook equations is mainly due to differences in the shape correcting factor ($A_r$) to find the optimal relation between drag coefficient (or equally said, Davies number; $X$) and Reynolds number ($Re$). This is the main parameterization error in this set of equations.

**2.2 Geometric and Fall Speed Measurements**

One source of uncertainty in applying the Böhm or Heymsfield and Westbrook (HW) method is calculating the area ratio ($A_r$) using instruments such as 2DVD or precipitation instrument package (PIP) as they do not give the projected area normal to the flow (i.e., they do not give the needed top view rather the 2DVD gives two side views on orthogonal planes as illustrated in Fig. 1). This is reasonable for snow aggregates which are expected to be randomly oriented. The other source

of uncertainty is in the definition of characteristic dimension used in $Re$ which in the HW method is taken to be the diameter of the circumscribing circle that completely encloses the projected area, the maximum dimension ($D_{max}$; this is what we use for the 2DVD in our application of the HW method). For the Böhm method we use the procedure in Huang et al. (2015) which used $D_{app}$ defined as the equal-volume spherical diameter.

The two-dimensional video disdrometer (2DVD) used herein is described in Schönhuber et al. (2000) and calibration and accuracy of the instrument are detailed in Bernauer et al. (2015). The 2DVD is equipped with two line-scan cameras (referred to as camera A and B) which can capture the particle image projection in two orthogonal planes (two side views). As mentioned earlier the area ratio ($A_r$) should be obtained from the projected image in the plane normal to the flow i.e., (top



or bottom view). However, to the best of our knowledge, there are no ground-based instruments that can automatically and continuously capture the horizontal projected views (i.e., in the plane orthogonal to the flow) of precipitation particles (however, 3D-reconstruction based on multiple views can give this information; Kleinkort et al. 2016). Compared with other optical-based instruments, such as HVSD (Hydrometeor Velocity Size Detector; Barthazy et al. 2004) or SVI (Snow Video

Imager; Newman et al. 2009) which only captures the projected view in one plane, the 2DVD offers views in two orthogonal planes giving more geometric information. Figure 1 shows a snowflake observed by a 2DVD from two cameras. The thick black line is the contour of the particle and the thin black lines show the holes inside the particle. The effective projected area $A_e$ in the definition of area ratio is easy to compute by counting total pixels from the particle's image, and then multiplying by horizontal and vertical pixel-width. The blue line is the minimum circumscribed ellipse. The area of the

ellipse is $A$ in the definition of area ratio. The size of particle measured by 2DVD is called apparent diameter ($D_{app}$) which is defined as the diameter of the equivalent volume sphere (Schönhuber et al. 2000; Huang et al. 2015). The $D_{app}$ is used when computing $Re$ as mentioned earlier. The area ratio and $D_{app}$ are the geometric parameters that are used in our implementation of the Böhm method.

In our application of the HW method the $A$ is based on the diameter of the circumscribed circle that completely encloses the projected pixel area ($A_e$) which is easy to calculate from the contours in Fig. 1. Thus the area ratio is $A_e/A$ while the characteristic dimension in $Re$ is the diameter of the circumscribing circle. Note that the area ratio and characteristic dimension in $Re$ depend on the type of instrument used (e.g., advanced version of snow video imager by von Lerber et al. 2017; the HVSD by Szyrmer and Zawadzki 2010). These instruments give projected view in one plane only and thus

geometric corrections are used as detailed in the two references.

The two optic planes of the 2DVD are separated by around 6 mm and the accurate distance is based on calibration by dropping 10 mm steel balls at three corners of the sensing area (details of the calibration as well as accuracy of size, fall speed, and other geometric measures are given in Bernauer et al. 2015). During certain time periods, more than one

precipitation particle falls in the 2DVD observation area. Since two cameras look at different directions, the particles observed by camera A and camera B need to be paired. This pairing procedure is called "matching", and it is illustrated in Fig. 2. The time period [$t_1$, $t_2$] is dependent on the assumed reasonable fall speed range. Assuming that the minimum and maximum reasonable fall speeds are $v_{min}$ and $v_{max}$, respectively, the distance between two optic planes is $D_d$, and that camera A observed a particle at $t_0$, we have $t_1 = t_0 + D_d/v_{max}$ and $t_2 = t_0 + D_d/v_{min}$. After matching, the fall speed can be calculated as

$D_d/\Delta t$, where $\Delta t$ is the time difference between two cameras observing the same particle. Because the fall speed of the 2DVD is dependent on matching, the geometric features and fall speeds will be in error when miss-match occurs. Huang et al. (2010) analyzed snow data from the 2DVD and found that the 2DVD manufacturer's matching algorithm for snow resulted in significant miss-matching problem (see, also, Bernauer et al. 2015). In the Appendix of Huang et al. (2010), they showed that miss-match will cause the volume, vertical dimension, and fall speed of particles to be over-estimated.





Subsequently, the mass of particles will also be over-estimated mainly because of fall speed. To get the best estimation of mass, they used 2DVD single camera data (before applying the manufacture's matching algorithm) and re-did matching based on a weighted Hanesch criteria (Hanesch 1999). The disadvantage of using single camera data, as in Huang et al. (2015), is that the particles' contour data is not available (i.e., the manufacturer's code does not provide line scan data from

5   single camera). Without contour data, both $D_{app}$ and $A$ can only be estimated by maximum width of the scan line and height of the particle as detailed in Huang et al. (2015). Moreover, the diameter of the circumscribing circle or ellipse cannot be obtained without contour data. The only quantity included in single camera data is $A_e$ in terms of number of pixels. The Huang, Bringi et al. (2015) approach is referred to as HB because both PSD (particle size distribution) and reflectivity ($Z_e$ ) are computed using $D_{app}$ as the measure of particle size.

In order to obtain contour data, we have to use the manufacturer's matching algorithm. To avoid over-estimating mass due to miss-match, we need to filter out those particles with unreasonable $v_t$. The vertical dimension of the particle's image before match is expressed as a number of scan lines (i.e., how many scan lines are masked by the particle). After match (so $v_t$ is known), the vertical pixel width is $v_t/f_s$, where $f_s$ is the scan frequency of a camera (~55 kHz), and the vertical size of the

particle is the vertical pixel width multiplied by the number of scan lines. Because two optical planes of the 2DVD are parallel, theoretically, the number of scan lines from cameras A and B should be the same. Considering the distance from the particle to the two cameras (projective effect of a camera), digital error of a camera and particle rotation in two planes, the difference in the number of scan lines between two cameras may not always be the same but should be very close. Hanesch (1999) gave a set of criteria for matching, the most important being the tolerance of the number of scan lines

between the two cameras (see Table 1). To obtain reliable fall speeds, we examined all matched particles (given by the manufacturer's matching algorithm) and removed those particles which did not satisfy the Hanesch scan line criteria. We used the fall speed of these filtered particles to compute their mass for both Böhm and Heymsfield-Westbrook methods, and then, dividing the mass by apparent volume (=$\pi D_{app}^3/6$), to get the particle density. Since the maximum density of ice particles is around 0.9 g cm$^{-3}$, we further remove particles whose density comes out as larger than 1 g cm$^{-3}$. After this two-

step filtering, the particles we use for further analysis are shown in Fig. 3 as blue points. The filtering will eliminate particles which will reduce the liquid equivalent snow accumulation. Hence the Pluvio gage accumulation is used as an integral constraint, i.e., the concentration in each bin is increased by a constant factor to match the 2DVD accumulation to the Pluvio accumulation.

### 2.3 Scattering Model

The scattering computation of ice particles is difficult because of their irregular shapes with large natural variability of shapes (e.g., snow aggregates or rimed crystals). The most common scattering method used in the meteorological community is the Discrete Dipole Approximation (DDA; Draine and Flatau 1994). However, DDA is very time consuming and not suitable for large numbers of particles especially at W-band (e.g., Chobanyan et al. 2015). On the other hand, the T-matrix



method (Mishchenko et al. 2002) is more time efficient and commonly used in radar meteorology but it requires that the irregular particle shape be simplified to an axis-symmetric shape (e.g., spheroid). Ryzhkov et al. (1998) have shown that, in Rayleigh region, the radar cross-section is mainly related to particle's mass-squared and less to the shape. For Mie scattering, however, the irregular snow shape plays a more significant role. Here we use two scattering models, one based on the soft

spheroid (Huang et al. 2015) with fixed axis ratio and quasi-random orientation. The apparent density is calculated as the ratio of mass to apparent volume. There is considerable controversy in the literature on the applicability of the soft spheroid model with fixed axis ratio especially at Ka and higher frequencies such as W-band (e.g., Petty and Huang 2010; Botta et al. 2010; Leinonen et al. 2012; Kneifel et al. 2015)). However, Falconi et al. (2018) used the soft spheroid scattering model using T-matrix to compute $Z_e$ (at X, Ka and W-bands) and showed that an effective optimized axis ratio of (oblate) spheroid

could be selected that matches directly measured $Z_e$ by radar (their optimal axis ratio, however, varied with the frequency band, i.e., 1 for X-band, 0.8 for Ka and 0.6 for W). They also found some differences in the optimal axis ratios for fluffy snow versus rimed snow. Nevertheless, they compared DDA calculations of complex-shaped aggregates to the soft spheroid model at W-band and concluded that the axis ratio can be used as a 'tuning' parameter. They also showed the importance of size integration to compute $Z_e$, i.e., the product of $N(D)$ and the radar cross-section for the soft spheroid versus complex-

shape aggregates. Their result implied that smaller particles had a larger value for the product when using soft spheroid of 0.6 axis ratio relative to complex aggregates and vice-versa for larger particles leading to a compensation when $Z_e$ is computed by size integration over all sizes. Thus, the soft spheroid model with axis ratio at 0.8 used by Huang et al. (2015) and which is used herein at Ku- and Ka-bands is a reasonable approximation.

The second scattering model we used herein is from Liao et al. (2013) who use an 'effective' fixed density approach to justify the oblate spheroid model. To compare the scattering properties of a snow aggregate with its simplified equal-mass spheroid, Liao et al. (2013) used 6-branch bullet rosette snow crystals with a maximum dimension of 200 □m and 400 □m, respectively, as two basic elements to simulate snow aggregation. They computed the backscattering coefficient, extinction coefficient, and asymmetry factor for simulated snowflakes, by the DDA, and for the corresponding spheres and spheroid

with the same mass but density fixed at 0.2 or 0.3 g cm⁻³, and hence the "apparent" sphere volume equals the mass divided by the assumed fixed density. They showed that when the frequency was lower than 35 GHz (Ka-band), the Mie scattering properties of spheres with a fixed density equal to 0.2 g cm⁻³ were in a good agreement with the scattering results for the simulated complex-shaped aggregate model with the same mass using the DDA (see, also, Kuo et al. 2016). They also showed this agreement with a spheroid model with a fixed axis ratio of 0.6 and random orientation. Here, we use the Liao et

al. (2013) equivalent spheroid model with a fixed 'effective' density of 0.2 g cm⁻³ at Ku- and Ka-bands (note that we estimate the mass of each particle from 2DVD measurements as described in Section 2.1). Note that this fixed-density spheroid scattering model is not based on microphysics (where the density would fall off inversely with increasing size) but on scattering equivalence with a simulated (same-mass) complex-shaped aggregate snowflake (Liao et al. 2016).



## 3 Case Analysis

### 3.1 Test Site Instrumentation and the Synoptic Event

The GPM Cold-season Precipitation Experiment (GCPEx) was conducted by the National Aeronautics and Space Administration (NASA), U.S.A., in cooperation with Environment Canada in Ontario, Canada from 17 January to 29
February, 2012. The goal of GCPEx was "… to characterize the ability of multi-frequency active and passive microwave sensors to detect and estimate falling snow …" (Skolfronik-Jackson et al. 2015). The field experiment sites were located north of Toronto, Canada between Lake Huron and Lake Ontario. The GCPEx had five test sites, namely, CARE (Centre for Atmospheric Research Experiments), Sky Dive, Steam Show, Bob Morton, and Huronia. The locations of five sites are shown in Fig. 4. The CARE site was the main test site for the experiment, located at $44^o13'58.44"$ N, $79^o46'53.28"$ W and
equipped with extensive suite of ground instruments. The 2DVD (SN37) and OTT Pluvio$^2$ 400 used for observations and analyses in this paper were installed inside a DFIR (Double Fence Intercomparison Reference) wind shield. The dual-frequency, dual-polarized Doppler Radar (D3R) was also located at the CARE site (Vega et al. 2014) near the 2DVD. The instruments used in this paper are depicted in Fig. 5. Because the radar and the instrumented site were nearly collocated, we can effectively view the set up as similar to 'vertical pointing' radar as described in more detail in Section 3.2.

We examine a snowfall event on 30-31 January 2012 that occurred across the GCPEx study area between roughly 22 UTC 30 January through 04 UTC 31 January. Details of this case using King City radar and aircraft spiral descent over the CARE site is given in Skolfronik-Jackson et al. (2015). This event resulted in liquid accumulations of roughly 1-4 mm across the GCPEx domain with fairly uniform snowfall rates throughout the event. At the CARE site the accumulations over an 8 h
period were <3.5 mm. Echo tops as measured by high altitude airborne radar was 7-8 km. The precipitation was driven by a shortwave trough moving from southwest to northeast across the domain. Figure 6 displays the 850 hPa geopotential heights (m), temperature (K), relative humidity (%), and winds (m s$^{-1}$) at 00 UTC 31 January, during the middle of the accumulating snowfall. A trough axis is apparent just to the west of the GCPEx domain (green star in Fig. 6). Low level warm air advection forcing upward motion is coincident with high relative humidity on the leading edge of the trough, over the
GCPEx domain (Fig. 6). Temperatures in this layer were around -10 to -15 °C throughout the event, supporting efficient crystal growth, aggregation, and potentially less dense snowfall as this is in the dendritic crystal temperature zone (e.g., Magono and Lee 1966). Aircraft probe data during a descent over the CARE site between 23:15-23:43 UTC showed the median volume diameter ($D_0$) of 3 mm, with particles up to maximum of 8 mm (aggregates of dendrites) at 2.2 km MSL with a large concentration of smaller sizes < 0.5 mm (dendritic and irregular shapes; Skofronik-Jackson et al. 2014). At the
surface, photographs of the precipitation types by the University of Manitoba showed small irregular particles and aggregates (< 3 mm) at 2330 UTC on 30 January.



### 3.2 D3R Radar Data

The D3R is a Ku- and Ka-band dual-wavelength, polarimetric scanning radar. It was designed for ground validation of rain and falling snow from GPM satellite-borne DPR (dual-frequency precipitation radar). The two frequencies used in the D3R are 13.91 GHz (Ku) and 35.56 GHz (Ka). These two frequencies were used for scattering computations in this research as

well. Some parameters of the D3R radar relevant for this paper are shown in Table 2. The range resolution of the radar is adjustable but usually set to 150 m and the near field distance is ~300 m; the practical minimum operational range is around 450 m. The minimum detectable signal of the D3R is −10 dBZ at 15 km. This means that when $Z_h$ is −10 dBZ at 15 km, the Signal to Noise Ratio (SNR) is 0 dB. Therefore, the SNR at any range, $r$, can be computed as:

$$SNR(r) = Z_h(r) + 10 + 20log_{10}\left(\frac{15}{r}\right) \text{ [dB] .} \tag{1}$$

The SNR is a very important indicator for radar data Quality Control (QC); the other important parameter for QC (in terms of detecting 'meteo' versus 'non-meteo' echoes) being the texture of the standard deviation (*std*) of the differential propagation phase ($\phi_{dp}$). We randomly selected 20 out of 85 RHI sweeps from 31 January, 2012 and computed the *std* of Ku-band $\phi_{dp}$ for each beam over 10 consecutive gates where SNR ≥ 10 dB. According to the histogram of the *std* of $\phi_{dp}$, 90% of the values were less than around 8°. Radar data at a range gate $m$ is identified as a 'good' data (i.e., meteorological

echoes) only if the standard deviation of $\phi_{dp}$ from the $(m − 5)^{th}$ gate to the $(m + 4)^{th}$ gate is less than 8°. This criterion sets a 'good' data mask for each beam at Ku-band. On the other hand, the $\phi_{dp}$ at Ka-band was determined to be too noisy and hence not used herein. The 'good' data mask for the Ka-band beam is set by the mask determined by the Ku-band criteria, with the additional requirement that the Ka-band SNR > 3 dB for the range gate to be considered as 'good'. Note that both radars are mounted on a common pedestal so that the Ku and Ka-band beams are perfectly aligned.

There are four scan types that can be performed by the D3R, namely, PPI (Plan Position Indicator), RHI (Range Height Indicator), surveillance, and vertical pointing. Figure 7 shows the scan strategies of the D3R on 31$^{st}$ January, 2012, which consisted of a fast PPI scan (surveillance scan; 10° per second) followed by four RHI scans (1° per second) except from 01:00 UTC to 02:00 UTC. The RHI scans with an azimuth angle of 139.9° point to the Steam Show site and those at 87.8°

point to the Sky Dive site. There were no RHI scans pointing to the Bob Morton site, and Huronia (52 km) was beyond the operational range (maximum 30 km) of the D3R. During the most intense snowfall the D3R scans did not cover the instrument clusters at Sky Dive and Steam Show sites. So we were left with analysis of D3R radar data at close proximity to the 2DVD or 'effectively vertical pointing' equivalent using RHI data from 75 to 90° at the nearest practical range of 600 m. PPI scan data at low elevation angle (3°) were also used from range gate at 600 m. The assumption is that there is little

evolution of particle microphysics from about 600 m height to surface and that the synoptic scale snowfall was uniform in azimuth (confirmed by Skolfronik-Jackson et al. 2015). The snowfall was spatially uniform around the CARE site so we selected data at 600 m range to compare with the 2DVD and Pluvio observations (this range was selected based on the





minimum operational range of 450 m; see Table 2) to which 150 m was added based on close examination of data quality. For RHI scans, the $Z_h$ at each band was averaged over the beams from 75° to 90°. The 75° is obtained from 600*cos(75°) ≈ 155 m which is close to the range resolution. For the fast PPI scan, $Z_h$ was averaged over all azimuthal beams at 600 m range.

Figure 8 shows the time profile of the averaged $Z_e$ at Ku- and Ka-bands. There are two problems indicated in this figure. First, theoretically, the Ku-band $Z_e$ should be greater than or equal to the Ka-band $Z_e$. The smaller Ku-band $Z_h$ indicates that a $Z$ offset exists at both bands. The other problem is that, compared with the Ka-band, there are many dips in the Ku-band $Z_h$. Comparing Fig. 8 with Fig. 7, we found that these dips occur only at RHI scans with azimuth angle larger than 300°. We

examined those RHI scans beam by beam from 90° to 75°. We further found that when the elevation angle is smaller than 78°, the unreasonably low $Z_h$ disappears. Therefore, the RHI scans with azimuth angles larger than 300° were averaged over the 75° to 78° elevation angles. To compute the *DWR*, we need to know the *Z* offset between the two bands. The measured $Z_h$ includes three components (neglecting attenuation):

$$Z_h^{meas} = Z_h^{true} \pm error(Z_h) + Z_{offset} \,, \tag{2}$$

where *error* refers to measurement fluctuations (typically with standard deviation of ~ 1 dB). The *DWR* is obtained as the difference between Ku-band $Z_h$ and Ka-band $Z_h$, with $Z_h$ being in units of dBZ. The measured *DWR* is:

$$DWR^{meas} = DWR^{true} \mp error(DWR) + \Delta Z_{offset} \,, \tag{3}$$

where error (*DWR*) is now increased since the Ku and Ka-band measurement fluctuations are uncorrelated (standard deviation of around 1.4 dB). The $\Delta Z_{offset}$ is determined by selecting data where the scatterers (snow particles) are

20 sufficiently small in size so that Rayleigh scattering is satisfied at both bands, i.e., $DWR^{true} = 0 \ dB$. The criteria used here is to select gates where Ku-band $Z_h < 0$ dBZ along with spatial averaging to reduce the measurement fluctuations in DWR to estimate $\Delta Z_{offset}$ in Eq. (3). Figure 9 shows the averaged $Z_h$ for the two bands from 20 RHI scans which satisfy the conditions above. After removing three extreme values (outliers) from Fig. 9, $\Delta Z_{offset}$ was estimated as −1.5 dB which is used in the subsequent data processing.

**3.3 2DVD Data Analysis**

The 2DVD used in this study was also located at the CARE site. The particle-by-particle mass estimation is based on three methods, respectively, as follows:

1)  Following the procedure in Huang et al. (2015) use 2DVD single-camera data and apply the weighted Hanesh matching algorithm (Hanesh 1999) to re-match snowflakes and PSD adjustment factor is computed by comparing the amout of

matched particles with single camera observed particles (details refer to Huang et al. 2015). Compute mass from fall speed, $D_{app}$ and environmental conditions using Böhm (1989). The 'apparent' density of the snow (ρ) is defined as


$6m/\pi D_{app}{}^3$. A mean power-law relation of the form $\rho = \alpha D_{app}{}^\beta$ is derived for the entire event as in Huang et al. (2015) as well as 1-min averaged $N(D_{app})$ is calculated. Note that the scattering model is based on the soft spheroid model with fixed axis ratio=0.8 and apparent density $\rho$. The results obtained by this method are denoted as "HB" method in the figures and in the rest of the paper.

2)   Use the manufacturer's (Joanneum Research, Graz, Austria) matching algorithm and filter miss-matched snowflakes as described in Section 2.2. The mass is computed from Böhm's equations. Following Liao et al. (2013) as far as the scattering model is concerned, the density is fixed at 0.2 g/cc and the volume is computed from mass=density*volume. The effective equal-volume diameter is $D_{eff}$ and the corresponding PSD is denoted as $N(D_{eff})$ which is different from $N(D_{app})$ in 1) above. Henceforth, this method is denoted as "LM".

3)   Use Joanneum matching and filtering method as in 2) but compute mass using Heymsfield-Westbrook equations as well as revised $D_{eff}$ and $N(D_{eff})$. This method is denoted as "HW". Thus, the only difference with 2) is in the estimation of mass and the difference in $D_{eff}$ and $N(D_{eff})$. The scattering model follows Liao et al. (2013).

The 2DVD measured liquid equivalent snow rate (SR) can be computed directly from mass as:

$$SR = \frac{3600}{\Delta t} \sum_{i=1}^{N} \sum_{j=1}^{M} \frac{V_j}{A_j}; \quad [mm\ hr^{-1}],\qquad\qquad (4)$$

where $\Delta t$ is the integral time (typically 60 s), $N$ is the number of size bins (typically 101 for the 2DVD), $M$ is the number of snowflakes in the $i^{th}$ size bin, and $A_j$ is the measured area of the $j^{th}$ snowflake. Further, $V_j$ is the liquid equivalent volume of the $j^{th}$ snowflake, so it is directly related to the mass. Figure 10 compares the liquid equivalent accumulation computed using the three methods above based on 2DVD measurements with the accumulation directly measured by the collocated Pluvio

snow gauge. The Pluvio-based accumulation at the end of the event (0330Z) was 1.9 mm while the 2DVD-measured accumulations using the three methods are 1.27 mm (HB), 1.45 mm (LM), and 1.24 mm (HW), respectively. It is expected that the PSDs of LM and HW should be under-estimated because of eliminating miss-matched particles which in principle could be re-matched. Re-matching of miss-matched particles is a research topic on its own and is beyond the scope of this paper. We used a simple way to adjust the PSD which is similar to the procedure in Huang et al. (2015) by scaling the PSD

by a constant so that the final accumulation from HB, LM and HW methods match the Pluvio data. Specifically, the PSD adjustment factors are 1.3 for LM and 1.52 for HW. Note that PSD adjustment of HB is not done by forcing 2DVD accumulated SR to agree with Pluvio. They are 1.54 for 00:00-00:45 UTC and 1.11 for 00:45-04:00 UTC. From the Pluvio data in Fig. 10 the SR is nearly constant at 0.7 mm/h between relative times of 1.5 to 3 h (or actual time from 01:00 on 30 Jan to 02:30 UTC).

The radar reflectivity at the two bands are simulated by using the T-matrix method assuming spheroid shape with axis ratio of 0.8 consistent with Falconi et al. (2018). The orientation angle distribution is assumed to be quasi-random with Gaussian



distribution for the zenith angle [mean=0°, σ=45°] and uniform distribution for the azimuth angle. However, other studies have assumed σ=10° (Falconi et al. 2018). The recent observations of snowflake orientation by Garret et al. (2015) indicate that substantial broadening of the snow orientation distribution can occur due to turbulence. Figure 11a,b compares the time series of D3R-measured $Z_h$ with the 2DVD-derived $Z_e$ for the entire event (2000-0330 UTC0 at (a) Ku and (b) Ka-band. The

5    $Z_e$ for both bands computed by the three methods generally agree with the D3R measurements to within 3-4 dB. Overall, LM gives the highest $Z_e$ and HB gives the lowest especially evident at Ka-band. This is consistent with scattering calculations by Kuo et al. (2016) of single spherical snow aggregates using constant density (0.3 g/cc) giving higher radar cross-sections and size-dependent density i.e., density falls of as inverse size (giving lower cross-sections). This feature is consistent with the scattering models referred to herein as LM and HB, respectively.

From 00:45 UTC to 01:30 UTC on 31 January 2012, the three 2DVD-derived $Z_e$ simulations deviate systematically from the D3R results for both bands. The other period is from 23:00 UTC to 23:30 UTC on 30 January 2012, when the Ku-band $Z_e$ has significant deviation from the D3R observations but the Ka-band $Z_e$ generally agrees with the D3R. Note that this synoptic event started at around 21:00 UTC on the 30[th] January and stopped at 03:30 UTC. We checked the D3R data and

found that before 22:30 UTC, the RHI scans were from 0° to 60°, so there were no usable data available for comparison with the 2DVD and Pluvio at the CARE site. We note that at 00:30 UTC the King City C-band radar recorded $Z_h$ in the range 15-20 dBZ around the CARE site which is in reasonable agreement with the D3R radar observations (Skolfronik-Jackson et al. 2015).

Figure 12a compares the time series of *DWR* simulated from 2DVD observations with the D3R measurements whereas Fig. 12b shows the scatterplot  In general, HB appears qualitatively in better agreement (better correlated and with significantly less bias) with D3R measurements relative to both LM and HW (significant underestimation relative to D3R). The scatterplot in Fig. 12b is an important result since in the HB method the soft spheroid scattering model is used with density varying approximately inverse with $D_{app}$ (density-$D_{app}$ power law where the larger snow particles have lower density).

Hence for a given mass the $D_{app}$ is larger (relative to Ka-band wavelength) and enters the Mie regime which lowers the radar cross-section at Ka-band (relative to same mass but constant density radar cross-section in LM and HW). Whereas at Ku-band the difference in radar cross-sections is less between the two methods (Rayleigh regime). The significant *DWR* bias in LM and HW relative to *DWR* observations is somewhat puzzling in that the Liao et al. (2013) scattering model radar cross-sections agree with the synthetic complex shaped snow aggregates of the same mass at Ka-band whereas the HB model

underestimates the radar cross-section relative to the synthetic complex shaped aggregates. On the other hand, Falconi et al. (2018) demonstrate that the soft spheroid model is adequate at X (close to Ku-band) and Ka-band and by inference adequate for *DWR* calculations with the caveat that different 'effective' axis ratios may need to be used at Ka- and W-bands.





We also refer to airborne (Ku,Ka) band radar data at 00:30 UTC which showed DWR measurements of 3-6 dB about 1 km height MSL around the CARE site but nearly 0 dB above that all the way to echo top (Skolfronik-Jackson et al. 2015). The latter is not consistent with aircraft spirals over the CARE site about an hour earlier where maximum snow sizes reach ~8 mm. In spite of the difficulty in reconciling the observations from the different sensors, the appropriate scattering model in

this particular event appears to favor the soft spheroid model used in HB based on better agreement with *DWR* observations. The other factor to be considered is the PSD adjustment factor which is assumed constant and independent of size which may not be the case, especially for the LM and HW methods as considerable filtering is involved due to mis-match (as discussed in Section 2.2). Note that a constant PSD adjustment factor will not affect *DWR* but it will affect $Z_e$. For the HB method Huang et al. (2015) determined the PSD adjustment factor for 4 events by comparing the 2DVD PSD to that

measured by a collocated SVI (snow video imager which was assumed to be 'truth') for each size bin. The PSD adjustment was found to be not size dependent for the HB method. On the other hand, because of the filtering of mis-matched particles by the LM and HW methods, the PSD adjustment factor maybe size dependent in which case the *DWR* will also change. More case studies are clearly needed to understand the applicability of the LM and HW methods of simulating *DWR*.

### 3.4 Snow Rate Estimation

To obtain radar-*SR* relationships, we use the 2DVD data and simulations. Since we employ a constant PSD adjustment factor and it will have similar influence on both $Z_e$ and *SR*, the 2DVD PSD used for scattering computation and SR need not be adjusted. Figure 13 shows the scatter plot of the 2DVD-derived $Z_e$ versus 2DVD-measured *SR* along with a power-law fit as $Z = a\ SR^b$. The fitting method used is based on Weighted Total Least Square (WTLS) so the power law can be inverted without any change. The coefficients and exponents of the power-law *Z-SR* relationship for both bands and three methods

are given in Table 3. It is obvious from Fig. 13 that there is considerable scatter at Ku-band for all three methods with the normalized standard deviation (NSTD) ranging from 55-70%. Whereas at Ka-band the scatter is significantly lower with NSTD from 40-45%. The errors in Table 3 are generally termed as parameterization errors.

By using dual-wavelength radar, we can estimate *SR* using $Z_e$ at two bands as:

$$\begin{cases} SR_{Ku} = a_1' * Z_{Ku}^{b_1'} \\ SR_{Ka} = a_2' * Z_{Ka}^{b_2'} \end{cases},$$
(5)

where $a' = (1/a)^{b'}$ and $b' = 1/b$. To reduce error, we may take the geometric mean of these two estimators as:

$$\overline{SR} = (SR_{Ku} * SR_{Ka})^{1/2} = c * Z_{Ku}^d * DWR^e ,$$
(6)

where $c = (a_1'a_2')^{1/2}$, $d = (b_1' + b_2')/2$, and $e = -b_2'/2$. Note that the *DWR* in Eq. (6) is in linear scale i.e., expressed as a ratio of reflectivity in units of mm$^6$ m$^{-3}$. Using Table 3 to set the initial guess of ($c,d,e$), non-linear least squares fitting was used

to determine the optimized (c,d,e) with the cost function being the squared difference between the 2DVD-based



measurements of *SR* and $c\, Z_{ku}{}^{d}\, DWR^{e}$, where $Z_{ku}$ and *DWR* are from 2DVD simulations. Figure 14 shows the *SR* computed from the 2DVD simulations of Ku-band $Z_e$ and the *DWR* using Eq. (6) versus the 2DVD-measured *SR*. The (c,d,e) values for the three methods are given in Table 4. As can also be seen from Fig. 14 and Table 4, the $SR(Z_{Ku},DWR)$ using LM method has the smallest NSTD (28.49%) but the other two methods have similar values of NSTD (≈30%). Although $SR(Z_{Ku},DWR)$ has smaller parameterization error than $Z_e$-*SR*, the $SR(Z_{Ku},DWR)$ estimation is biased high when *SR* < 0.2 mm/hr (see Fig. 14). When *SR* is small, the size of snowflakes is usually also small and falls in the Rayleigh region at both frequencies resulting in *DWR* very close to 1 (when expressed as a ratio). This implies that there is no information content in the *DWR* so including it just adds to the measurement error. Hence, for small *SR* or when DWR≈1, we use the $Z_e$-*SR* power law.

So far the single frequency *SR* retrieval algorithms were based on 2DVD-based simulations with a PSD adjustment factor using the total accumulation from Pluvio as a constraint. The algorithm we propose for radar-based estimation of *SR* is to use Eq. (6) when *DWR* > 1 and *SR* > 0.2 mm/h else we use the $Z_{Ka}$-*SR* power law (note: we don't use the $Z_{Ku}$-*SR* power law as the measurement errors of $Z_{Ku}$ seem to be on the high side, Fig. 9). The precise thresholds used herein are *ad hoc* and may need to be optimized using a much larger data set. Figure 15a shows the radar-derived accumulation using $Z_{Ka}$-*SR* versus the Pluvio accumulation versus time. The total accumulation from the Pluvio is 2.5 mm and the three radar-based total accumulations , respectively for HB, LM and HW methods amount to [2.6, 1.8, 2.6 mm]. Except for the underestimate in the LM method (-28 %) the other two methods agree with the Pluvio accumulation in this event. Figure 15b is same as Fig. 15a except the combination algorithm mentioned above is used  For this case, ~33% of data used the $Z_{Ka}$-*SR* power law due to threshold constraints given above. The event accumul*tions* , respectively for HB, LM and HW methods amount to [2.4, 1.9, 2.2 mm] consistent with the algorithm that uses only the $Z_{Ka}$-*SR* power law. However, the criteria of relative bias error in the total accumulation (in events with low accumulations such as this one) is not necessarily an indication that the DWR-based algorithms is not adding value. Rather, the criteria should be snow rate inter-comparison which could not be done due to the low resolution (0.01 mm/min) of the Pluvio[2] 400 gage along with the low event total accumulation of only 2.5 mm. A close qualitative examination of Fig. 15b shows that the HB method more closely 'follows' in time the gage accumulation relative to HB in Fig. 15a. In Fig. 15, the time grid is different for the radar-based data and the gage data. It is common to linearly interpolate the gage data to the radar sampling time and if this is done, the *rms* error for the HB method reduces from 0.1 mm (when using only the $Z_{Ka}$-*SR* power law) to 0.045 mm for the DWR algorithm, which constitutes a significant factor of two reduction.

The total error in the radar estimate of *SR* is composed of both parameterization errors as well as measurement errors with measurement errors dominating since the *DWR* involves the ratio of two uncorrelated variables. From Section 8.3 of Bringi and Chandrasekar (2001) the total error of *SR* in Eq. (6) is around 50% (ratio of standard deviation to the mean). The assumptions are, (a) the standard deviation of the measurement of $Z_e$ is 0.8 dB, (b) the standard deviation of the *DWR* (in dB)



measurement is 1.13 dB, and (c) the parameterization error is 30% from Table 4. However, considering the $Z_e$ fluctuations in Fig. 9, the measurement standard deviation probably exceeds 0.8 dB, especially at Ku-band. Thus, sufficient smoothing of *DWR* is needed to minimize as much as possible the measurement error at the same time maintaining sufficient spatial resolution.

**4 Summary and Conclusions**

The main objective of this paper is technique development for snow estimation using scanning dual-wavelength radar operating at Ku and Ka-bands (D3R radar operated by NASA). We use the 2D-video disdrometer and collocated Pluvio gage to derive an algorithm to retrieve snow rate from reflectivity measurements at the two frequencies as compared to the conventional single-frequency $Z_e$-*SR* power laws. The important microphysical information needed is provided by the 2DVD

to estimate the mass of each particle knowing the fall speed, apparent volume, area ratio and environmental factors from which an average density-size relation is derived (e.g., Huang et al. 2015; von Lerber et al. 2017; Bohm 1989; Heymsfield and Westbrook 2010).

We describe in detail the data processing of 2DVD camera images (in two orthogonal planes) and the role of particle mis-

15 matches that give erroneous fall speeds. We use the Huang et al. (2015) method of re-matching using single camera data but also use the manufacturer's matching code with substantial filtering of the mis-matched particles since the apparent volume and diameter ($D_{app}$) are more accurate. To account for the filtering of the mis-matched particles, the particle size distribution is adjusted by a constant factor using the total accumulation from the Pluvio as a constraint.

Two scattering models are used to compute the $Z_{Ku}$ and $Z_{Ka}$ termed as the soft spheroid model (Huang, Bringi et al. 2015; HB method) and the Liao-Meneghini (LM) model which uses the concept of "effective" density. In these two methods the particle mass is based on Böhm (1989). The method of Heymsfield and Westbrook (2010) is also used to estimate mass which is similar to Böhm (1989) but is expected to be more accurate (Westbrook and Sephton 2017); along with the LM model for scattering this method is termed as HW.

The case study chosen is a large scale synoptic snow event that occurred over the instrumented site. The $Z_{Ku}$ and $Z_{Ka}$ were simulated based on 2DVD data and the three methods i.e., HB, LM and HW yielded similar values within ±3 dB. When compared with D3R radar measurements extracted as a time series over the instrumented site, the LM and HW methods were closer to the radar measurements with HB method being lower by ≈3 dB. Some systematic deviations of simulated

reflectivities by the three methods from the radar measurements were explained by possible size dependence of the PSD adjustment factor.





The direct comparison of *DWR* (ratio of $Z_{Ku}$ to $Z_{Ka}$) from simulations with DWR measured by radar showed that the HB method gave the lowest bias with the data points more or less evenly distributed along the 1:1 line. The simulation of *DWR* by LM and HW methods underestimated the radar measurements of *DWR* quite substantially even though the correlation appeared to be reasonable. The reason for this discrepancy is difficult to explain since a constant PSD adjustment factor

(slightly different for the 3 methods) would not affect the *DWR*. From the scattering model viewpoint, the LM method takes into account the complex shapes of snow aggregates via an 'effective' density approach whereas the HB method uses soft spheroid model with density varying approximately inversely with size. We did not attempt to classify the particle types in this study.

The retrieval of *SR* was formulated as $SR = c * Z_{Ku}^{d} * DWR^{e}$ where [*c, d, e*] were obtained via non-linear least squares for the three methods. The total accumulation from the three methods using radar-measured $Z_{Ku}$ and *DWR* were compared with the total accumulation from the Pluvio (2.5 mm) to demonstrate closure. The closest to Pluvio was the HB method (2.4 mm), next was the HW method (2.24 mm) and LM gave (1.94 mm). At such low total accumulations, the three methods show good agreement with each other as well as with the Pluvio gage. The poor resolution of  the gage combined with the

relatively low total accumulation  in this event precluded direct comparison of snow rates. The combined estimate of parameterization and measurement errors for snow rate estimation was estimated around 50%. From variance decomposition, the measurement error variance as a fraction of the total error variance was 58% and the parameterization error variance fraction was 42%. Further, the *DWR* was responsible for 90% of the measurement error variance which is not surprising since it is the ratio of two uncorrelated reflectivities. Thus, the *DWR* radar data has to be smoothed spatially (in

range and azimuth) to reduce this error which will degrade the spatial resolution but is not expected to pose a problem in large scale synoptic snow events.

**Data Availability**

The data used in this study can be made available upon request to the corresponding author.

**Competing Interests**

The authors declare that they have no conflict of interest.

**Acknowledgements**

Author GWH acknowledges support by BrainPool 2017, Rep. of Korea.  All authors except GWL acknowledge support from NASA PMM Science grant NNX16AE43G.



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



**Tables**

**Table 1:** Hanesch 2DVD scan line criteria.

| Max. of Total Scan Lines | Difference of Scan Lines |
|:---:|:---:|
| ≤ 20 | < 3 |
| 21-44 | 15%-11% |
| 45-181 | 11% |
| ≥182 | 20 |





**Table 2:** Some D3R parameters relevant for this study. Full D3R specifications can be found in Chandrasekar et al. 2010.

|  | Ku | Ka |
|---|---|---|
| Frequency (GHz) | 13.91 | 35.56 |
| Min. Detectable Signal | -10 dBZ at 15 km | |
| Range (km) | 0.45 - 30 | |
| Range Resolution (m) | 150 | |
| Ant. Beam Width | $\sim 1^{o}$ | |




**Table 3:** Coefficients and exponents of the power-law *Z-SR* relationship for "HB", "LM", and "HW" methods and Ku- and Ka-bands, respectively.

| Method | Band | a | b | STD (mm/hr) | NSTD (%) |
|--------|------|------|------|-------------|----------|
| HB | Ku | 140.52 | 1.48 | 0.2156 | 70.99 |
| | Ka | 60.17 | 1.18 | 0.1366 | 44.97 |
| LM | Ku | 129.27 | 1.64 | 0.2235 | 55.89 |
| | Ka | 99.85 | 1.25 | 0.1614 | 40.35 |
| HW | Ku | 106.25 | 1.58 | 0.1889 | 55.30 |
| | Ka | 66.96 | 1.42 | 0.1473 | 43.11 |



**Table 4:** Coefficients and exponents of the $SR(Z_{Ku}, DWR)$ relation [see Eq. (10)] for three methods.

| Method | c | d | e | STD (mm/hr) | NSTD (%) |
|--------|------|------|------|-------------|----------|
| HB | 0.0632 | 0.6537 | -0.9155 | 0.0986 | 32.45 |
| LM | 0.0995 | 0.5648 | -1.3415 | 0.1139 | 28.49 |
| HW | 0.1017 | 0.5426 | -1.1772 | 0.1076 | 31.52 |





**Figures**

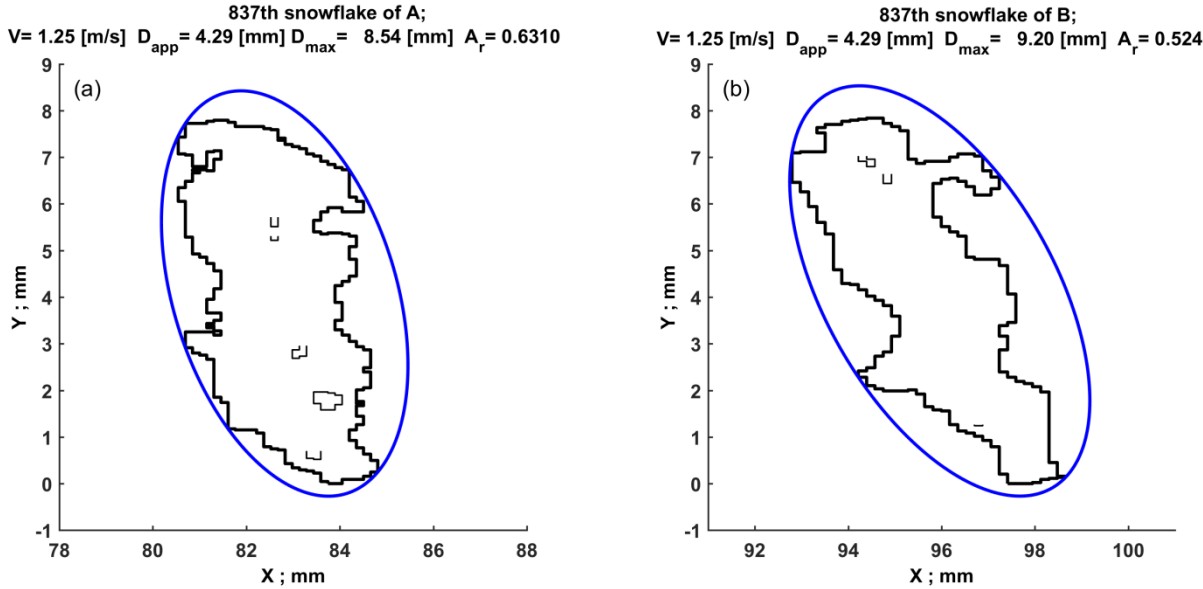

**Figure 1:** A snowflake observed by a 2DVD from two views. The thick black line is the contour of the snowflake and the thin black lines show the holes inside the snowflake. The effective area, $A_e$, equals the area enclosed by the thick black curve minus the area enclosed by thin lines. The blue line represents the minimum circumscribed ellipse, whose enclosed area is denoted by $A$.




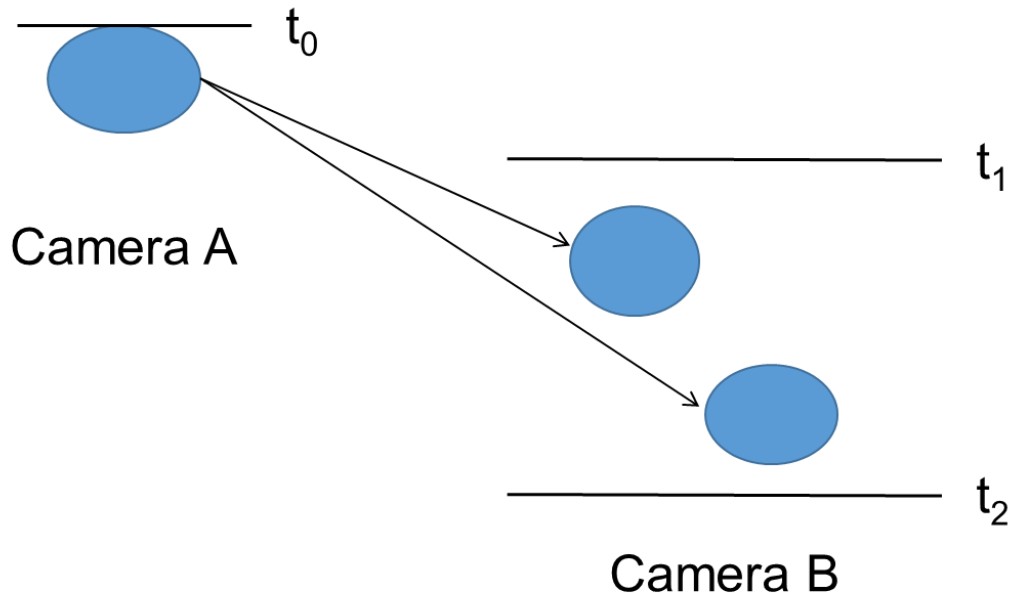

**Figure 2:** Illustration of the matching procedure. In the situation shown, it is assumed that camera A observed a particle at time $t_0$, and afterwards during a certain time period $t_1$ to $t_2$, camera B observed two particles. The matching procedure decides which particle observed by camera B is the same particle observed by camera A.





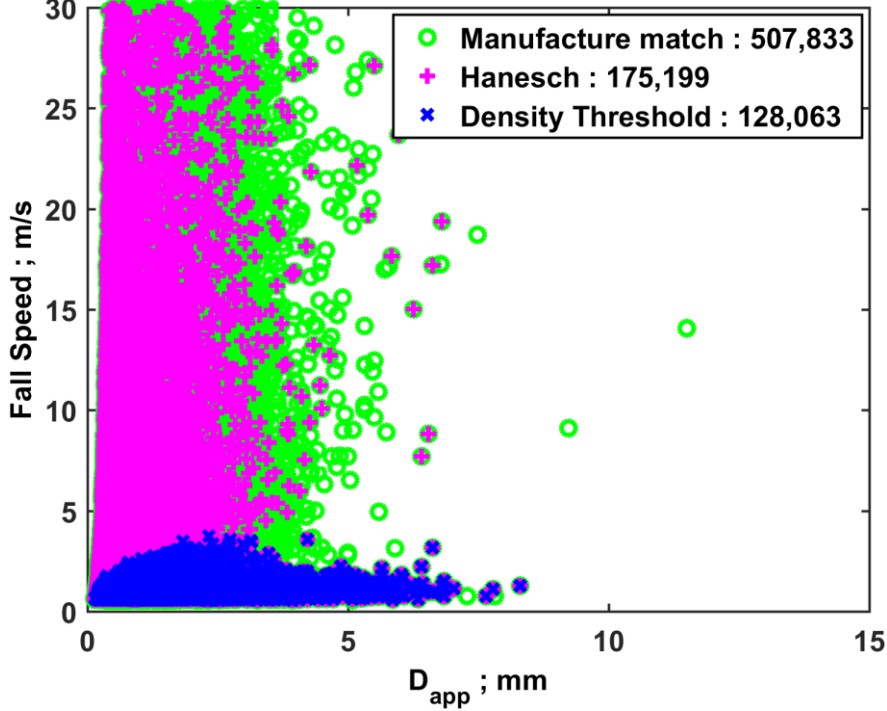

**Figure 3:** Fall speed versus $D_{app}$ for the synoptic case on January 31, 2012 at the CARE site. The green circles represent the results of the manufacturer's matching algorithm. The magenta crosses are the matched particles which satisfy Hanesch scan line criteria. The blue "x"s are the density of particles (mass computed by Böhm's equations) lower than 1 g cm$^{-3}$.





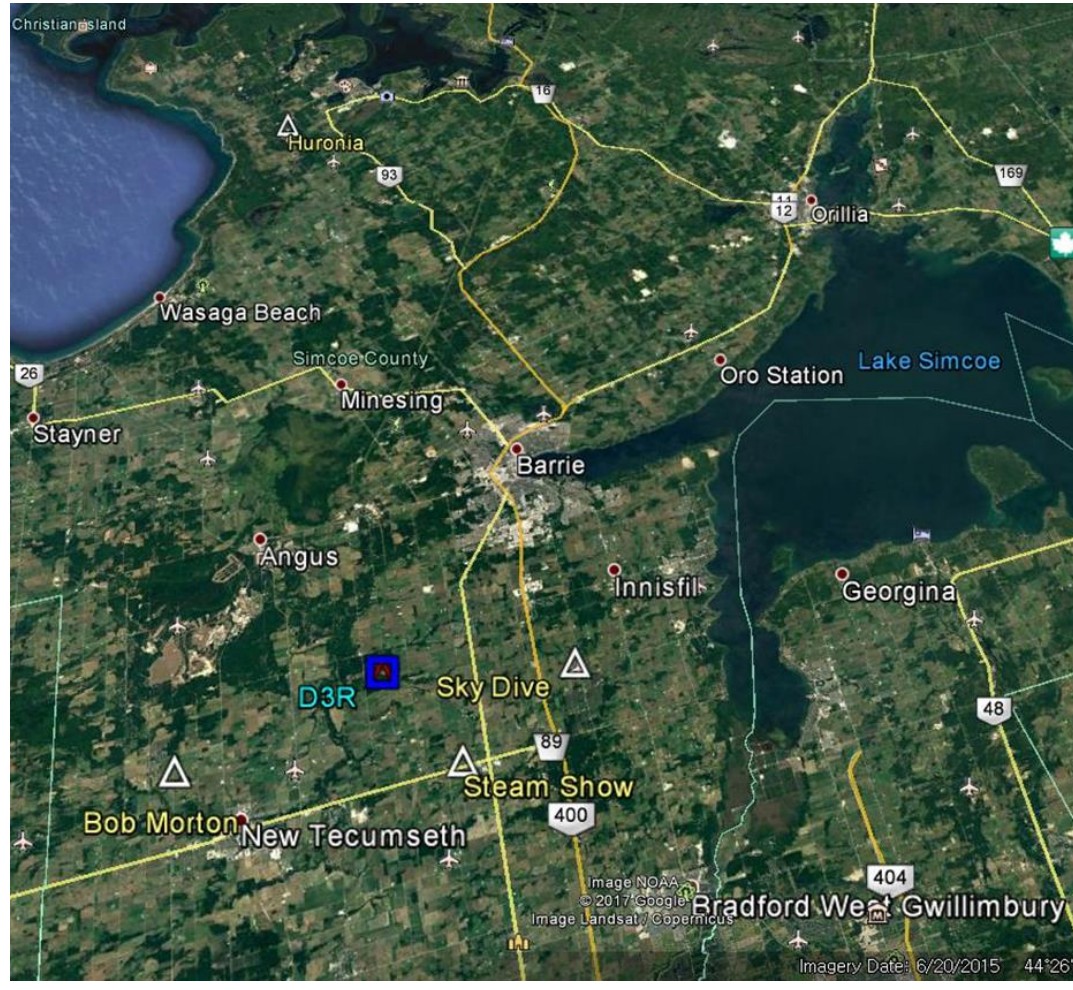

**Figure 4:** A map of the GCPEx field campaign. The five test sites are CARE, Sky Dive, Steam Show, Bob Morton, and Huronia. The ground observation instruments, namely, 2DVD, D3R, and Pluvio, used in this research were located at CARE.





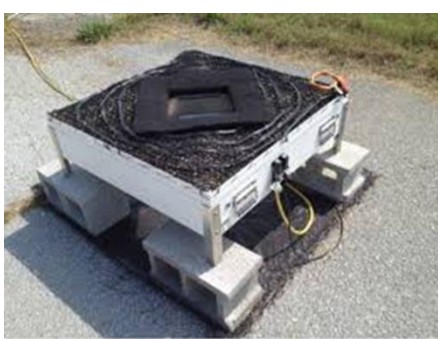
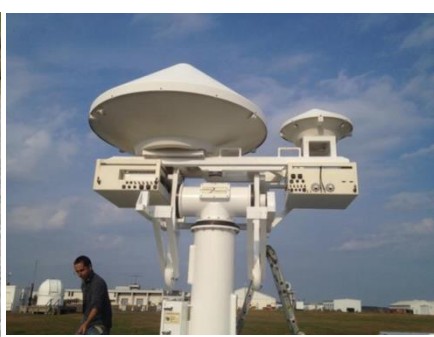
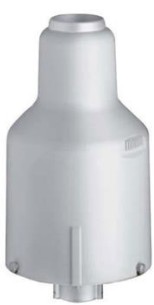

**Figure 5:** Instruments used in this study (from left to right): 2DVD (SN37), D3R (Dual-polarized Doppler Radar), and OTT Pluvio$^2$ 400 precipitation gauge.





**Figure 6:** The 00 UTC 31 January 2012 850 hPa geopotential heights (m, black solid contours), temperature (K, red - above freezing, blue –below freezing), relative humidity (%, green shaded contours), and wind (m s$^{-1}$, wind barbs). The red dot in the center right portion of the figure denotes the general location of the GCPEx field instruments.

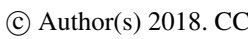



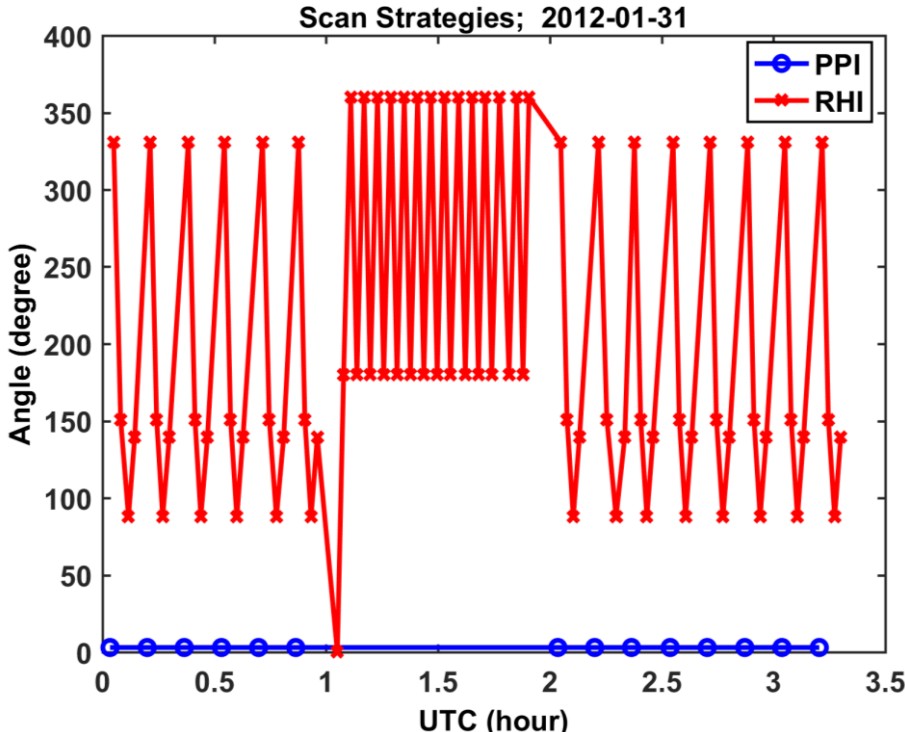

**Figure 7:** D3R scan strategies on 31 January 2012. The Y-axis is azimuth angle (RHI; red "x") or elevation angle (PPI; blue "o"). The scan rate of RHI was 1°/s and 10°/s for PPI.





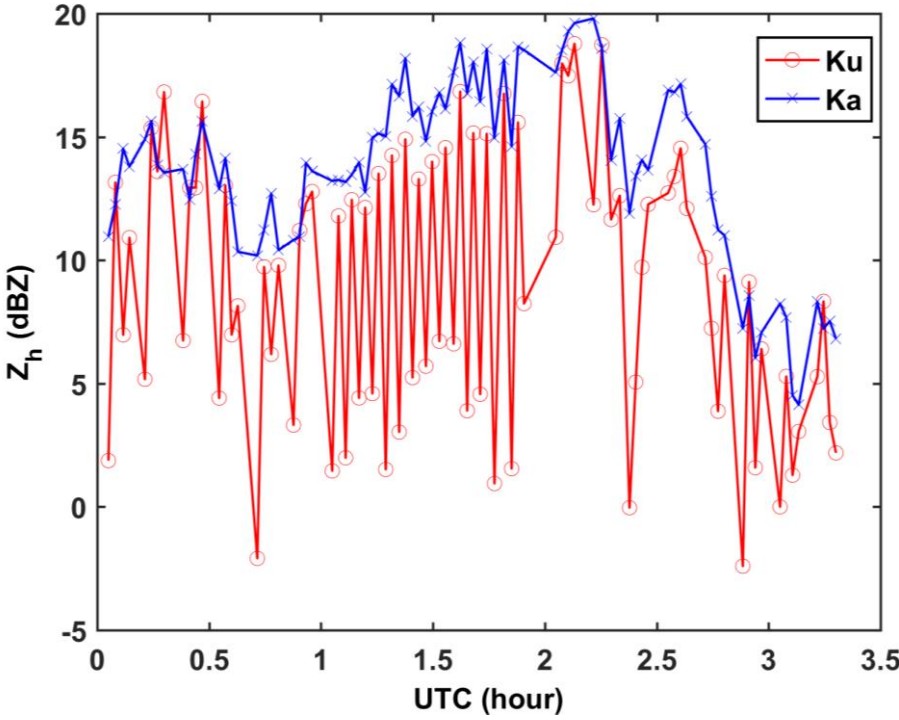

**Figure 8:** The time series of averaged raw $Z_h$ at the CARE site. There are two problems indicated in this figure: (i) The Ku-band $Z_h$ is smaller than the Ka-band $Z_h$ on average. (ii) Compared with the Ka-band, there are many too small values of the Ku-band $Z_h$.





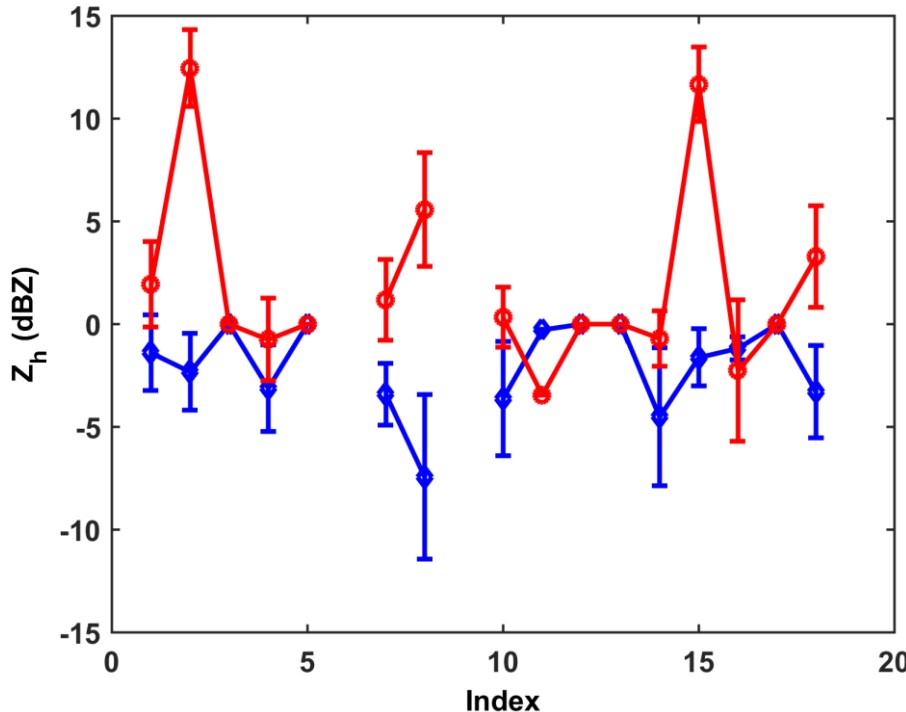

**Figure 9:** The averaged raw $Z_h$ for Ku- and Ka-bands. The $Z_h$ was randomly selected from 20 of 85 RHI scans with Ku-band $Z_h < 0$ dBZ, range < 1 km, and Ka-band SNR >3 dB.




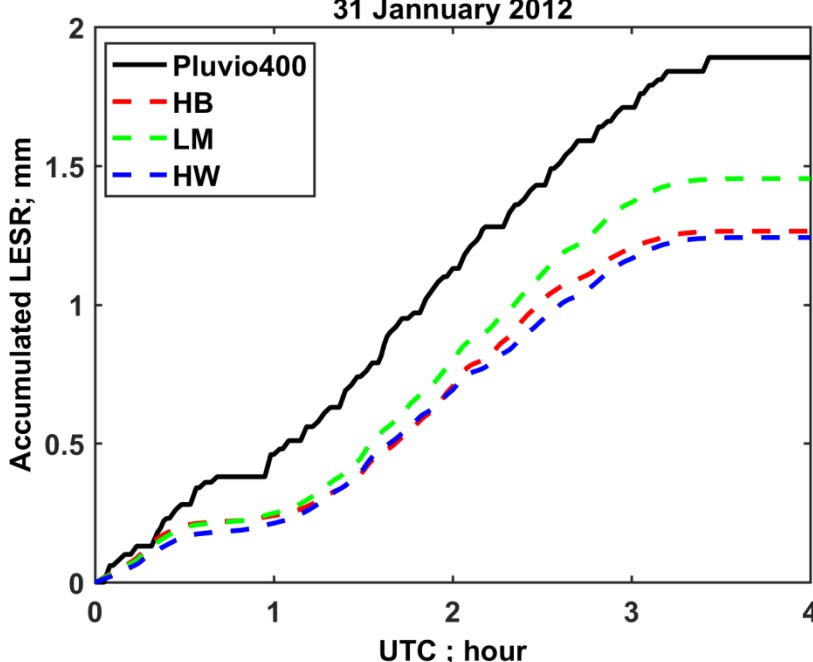

**Figure 10:** Comparison of liquid equivalent accumulations computed using HB, LM, and HW methods, respectively, based on 2DVD measurements and that directly measured by the collocated Pluvio snow gauge. We used the total accumulation to estimate the PSD adjustment factor for the "LM" and "HW" methods.





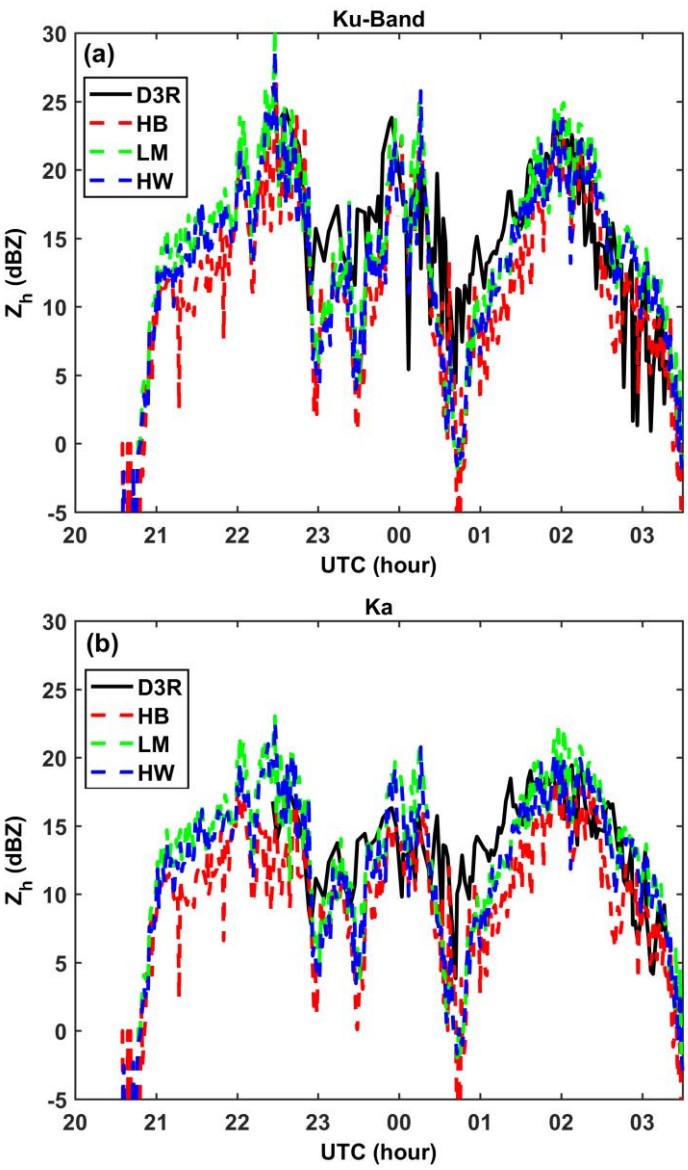

**Figure 11:** Comparison of the 2DVD derived $Z_h$ with D3R measurements for the entire event, for Ku-band (a) and Ka-band (b). This synoptic system started at around 2100Z on 30 January and ended at around 0330Z on 31 January 2012. $Z_e$ by "LM" is close to "HW" and slightly higher, whereas "HB" method gives the lowest $Z_e$. $Z_e$ results computed by all methods generally agree with D3R measured $Z_h$.





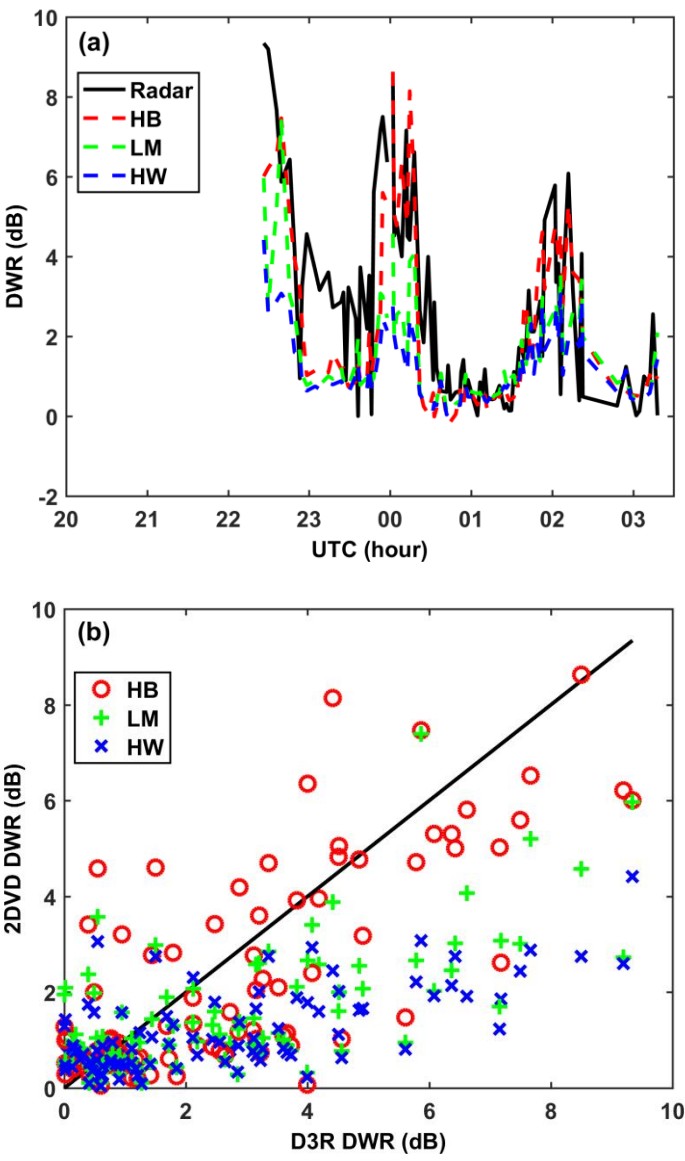

**Figure 12:** Comparison of the 2DVD derived DWR using "HB", "LM", and "HW" methods, respectively, with the D3R measured *DWR*. (a) shows the time profile of the D3R, and (b) shows the scatter plot.



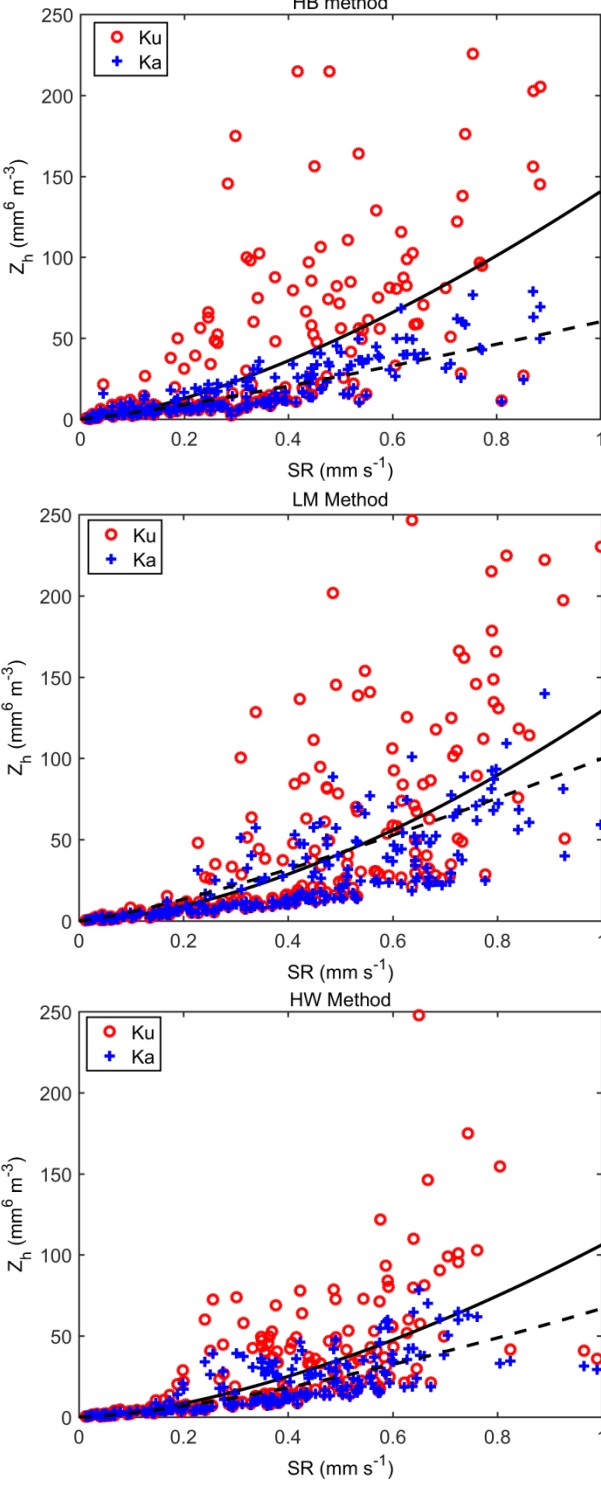

**Figure 13:** 2DVD derived $Z_h$ versus 2DVD measured *SR* scatter plots, with *Z-SR* power-law fits, for Ku- and Ka-bands and "HB" method (top panel), "LM" method (central panel), and "HW" method (bottom panel).





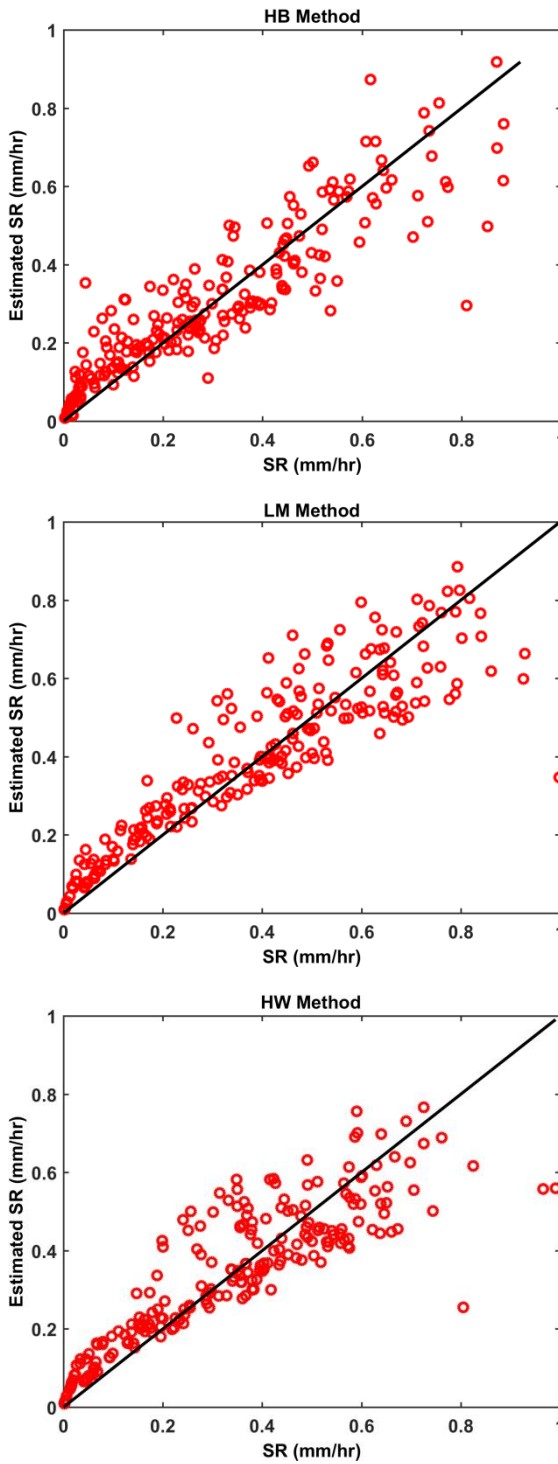

**Figure 14:** Estimated *SR* using $Z_e$ and DWR of the 2DVD and Eq. (10) versus 2DVD *SR* scatter plot, for "HB" method (top panel), "LM" method (central panel), and "HW" method (bottom panel).





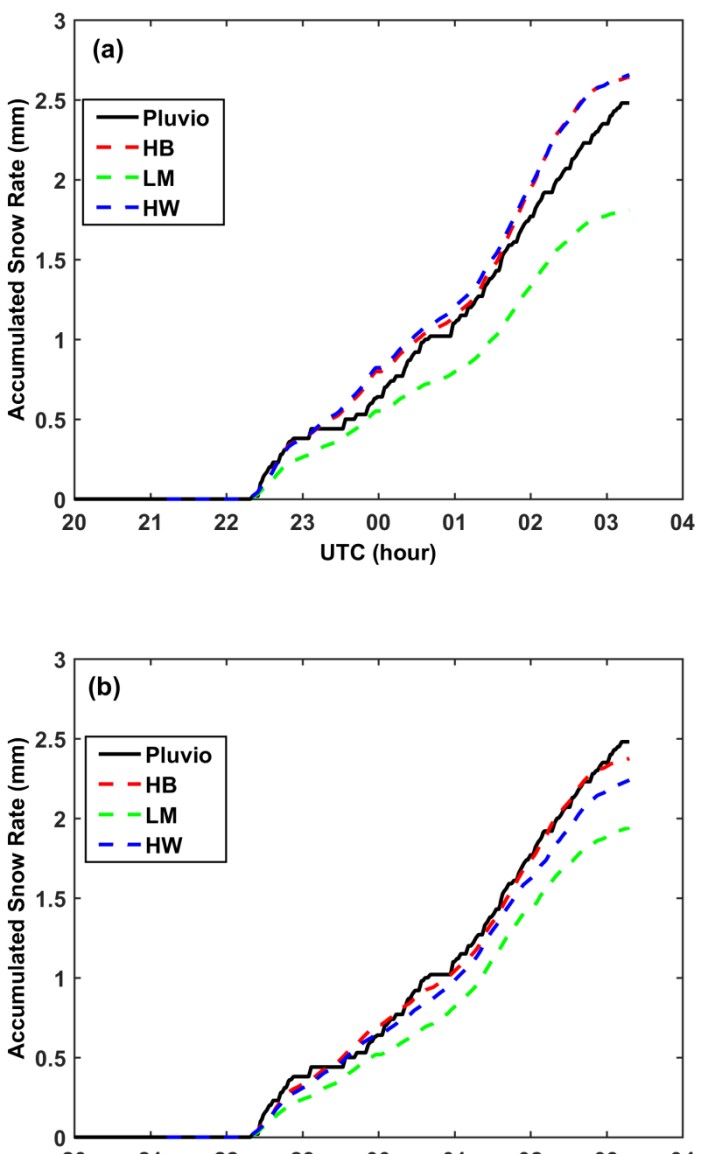

**Figure 15:** Comparison of the radar derived accumulated *SR* using "HB", "LM", and "HW" methods, respectively, with Pluvio gauge measurement. (a) The radar *SR* is computed by $Z_{Ka}$-*SR* relationships. The Pluvio accumulated *SR* on 03:18 UTC is 2.48 mm. The radar accumulated *SR*s for "HB", "LM", and "HW" are 2.64, 1.81 and 2.66 mm respectively. (b) The radar SR is computed by combining $SR(Z_{Ku}, DWR)$ and $Z_{Ka}$-*SR* as described in the text. The accumulated *SR* derived from the radar using "HB" method is 2.38 mm, using "LM" is 1.94 mm, and using "HW" is 2.24 mm.