# Peer review of "Dual-Wavelength Radar Technique Development for Snow Rate Estimation: A Case Study from GCPEX"

_Atmospheric Measurement Techniques, 2018_

## Referee Comment (RC1) · Anonymous Referee #1 · 3 Sep 2018

This article describe a dual-frequency method to estimate the amount of snowfall from radar measurements obtained by NASA's D3R radar during the GCPEX experiment in 2012. The proposed method hinges on a representation of the Z-SR relationship conditional to the DWR. The Authors demonstrate the superiority of their DWR-based algorithm when compared to traditional power-law relationships to retrieve the liquid-equivalent snow rate. The article provides a nice illustration of the use of in situ microphysical data, with radiative-transfer models (T matrix with various assumptions about the mass-size relationship) constrained by remote sensing observations. However, before I can recommend this article for publication, the Authors should revise a few key points detailed below. The writing is of unequal quality

with some paragraph extremely well written when others a filled with typos and unclear sentences. It would also be possible good to reduce the length of the manuscript by removing 1 or 2 figures and the overly long part that details the processing of the 2DVD images. Lastly, I couldn't find much about the efforts of the Authors to avoid or mitigate the effects of attenuation on the radar measurements, particularly at Ka band. Failing to do so can significantly bias the retrievals performed using the radar observations.

Please also note the supplement to this comment:
https://www.atmos-meas-tech-discuss.net/amt-2018-211/amt-2018-211-RC1-supplement.pdf

**Supplement:**

This article describes a dual-frequency method to estimate the amount of snowfall from radar measurements obtained by NASA's D3R radar during the GCPEX experiment in 2012. The proposed method hinges on a representation of the Z-SR relationship conditional to the DWR. The Authors demonstrate the superiority of their DWR-based algorithm when compared to traditional power-law relationships to retrieve the liquid-equivalent snow rate.

The article provides a nice illustration of the use of in situ microphysical data, with radiativetransfer models (T matrix with various assumptions about the mass-size relationship) constrained by remote sensing observations. However, before I can recommend this article for publication, the Authors should revise a few key points detailed below. The writing is of unequal quality with some paragraph extremely well written when others a filled with typos and unclear sentences. It would also be possible good to reduce the length of the manuscript by removing 1 or 2 figures and the overly long part that details the processing of the 2DVD images. Lastly, I couldn't find much about the efforts of the Authors to avoid or mitigate the effects of attenuation on the radar measurements, particularly at Ka band. Failing to do so can significantly bias the retrievals performed using the radar observations.

Detailed comments and suggestions (technical questions in italic)

- General comment
  - Please decide whether to use "gauge" (preferred) or "gage" and use this consistently throughout the manuscript. Similarly for mis-/miss-/mismatch
  - Please provide a table of acronyms and symbols.
- Introduction
  - Page 2 Line12 (P2L12): "In this study, we" ...
  - P2L29: Please introduce the "Dual-\lambda" notation (e.g. in line 26) before using it.
  - P3L4: "ratio of 4th moment to 3rd moment... ": "Dm" the mean mass-weighted diameter will be the ratio of the 4th to the 3rd moment only if the exponent of the m-D relationship is 3, which is seldom the case when dealing with snow... please reformulate.
  - P3L14: "estimate the mass of"
  - P3L16: "based on a particle's mass"
- Section 2
  - P3L29: "dominates"
  - P4L16: please include a space between "\delta\_0" and "in Bohn";
  - *P4L19: "parameterization error": Is this an error (i.e. producing wrong results) or a different convention?*
  - P5L15: "circumscrib<mark>eding</mark> circle"?
  - P5, last paragraph to the end of Section 2.2: This paragraph provides waaay too much detail on the operation of the 2DVD. Please refer to literature or move to an appendix. Similarly, I'm not sure that Figure 2 is really needed...
  - P5L21&28: "optical planes"?
  - P5L31 to 34: is it miss-match or mismatch?

- P6L2: "manufacturer's matching"
- *P6L8-10: "The Huang... because... size". I do not understand the causality in this sentence. Please rephrase.*
- P6L26: "particles, which ... Hence, the"
- P6L4: "Here, we use"
- P6L22: There is a typo in the units reported (200 and 400 \mu m ?)
- P6L24: "spheroids"
- Fig3: Since both quantities being plotted are positive, would a plot in log-log scale be more appropriate/revealing?

**- Section 3**

- P8L8-9: Is Figure 4 necessary? It is already in Skofronick et al 2015... Please consider deleting this Figure otherwise.
- P8L10: "with an extensive"
- P8L14: "similar to a 'vertical pointing"
- P8L20: "Echo tops... high-altitude radar was were"
- Compared to Section 3.1, Section 3.2. is really well written. The Authors provide a clear description of their thorough QC efforts of the DWR data.
- What do the Authors do to address attenuation especially at Ka band? This could have a strong effect on the DWR, right?
- P10L29: "amount of"
- P11L22: "particles which, in principle, could"
- P12L3: "Figure 11a,b compares"
- P12L8: "falls off as"
- *P13L1: For the airborne radar data, please refer to APR2 and cite appropriate references.*
- P13L2: "Skolfronik"?
- P13L7: "mis-match" please decide between "miss-match, mis-match and mismatch" for the entre article...
- P13L11: "found not to be not size dependent"
- P13L12: "factor may be size"
- P13L20: "It is obvious from Fig. 13 that there is considerable scatter", how about in a loq-loq scale, is there still a significant scatter?
- P13L21-22: ".. ranging from 55 to 70%... from 40 to 45%"
- P14L1: Please re-write all "Ku" or "Ka" with a capital "K" even in the subscripts
- P14L14: "don't do not use"
- P14L16-l17: There is a truly remarkable agreement between measured and simulated accumulated snowfall!
- P14L18: "Figure 15b is the same as..."
- P14L20: why italics in "accumulations"?
- P15L3: "at the same time while maintaining"
- Section 4

- P15L6: "is to develop a technique development for ... using a scanning"
- P15L26: "a larg<mark>e-s</mark>cale synoptic ... site of CARE during GCPEx."
- P15L30: "explained by a possible"
- P16 1st paragraph: Could the discrepancies that you couldn't explain be due to attenuation, which could strongly affect the Ka-band measurements?
- P16L13: "and LM gave (1.94 mm)."
- P16L19-21: Is the smoothing of DWR actually a smoothing of Ka or is it necessary to first form DWR and then smooth only this parameter?
- Could you also provide some perspective as to how to extend your method to other dual-frequency datasets? Would the methods presented here still work if applied to a Ku-W or Ka-W data set? What would be the expected differences/commonalities? Similarly, would there be some potential or added value for airborne or spaceborne dual-frequency radars? Lastly, how would you extend your method to handle more frequencies (3,4,..)?
- References
  - P17L24: "for scattering"
  - P17L31: "Orinetations"
  - 0
- Figures
  - Figure 10: legend: "that were directly measured..."
  - Figures 13 and 14:
    - Please use the same units for SR between the two plots (Figs 13 and 14) to help the reader compare plots;
    - In Fig. 13, would a logarithmic scale work better?

---

## Author Comment (AC1) · 29 Sep 2018

Author Comment on behalf of all Co-Authors (AC) constituting Authors' Response to Interactive comment from Anonymous Referee #1 is provided in the attached pdf document (supplement). We would like to thank the reviewer for the positive general comments and detailed technical and editorial comments (corrections, suggestions, and questions). We have responded in the attached pdf supplement to all of the comments (except those that are entirely complimentary) and included changes in the revised manuscript.

[Figure]

Please also note the supplement to this comment:
https://www.atmos-meas-tech-discuss.net/amt-2018-211/amt-2018-211-AC1-supplement.pdf
* * *

---

## Referee Comment (RC2) · Anonymous Referee #2 · 10 Oct 2018

A method is proposed to estimate snowfall rates from the D3R dual-frequency radar measurements. The method is developed and assessed during a snowfall event in 2012. 2D-video disdrometer and gauge data are used to link observed radar reflectivity and particles physical properties. The dual-frequency estimator is shown to estimate snowfall rates with more accuracy than the conventional single frequency Z-S approach.

The paper is well written. The detailed discussion on the assumptions, methodology and techniques is appreciated. Dual-frequency estimation of snowfall rates addresses the critical need for improved snow estimation from ground- and space-based remote

sensing. The work presented is worthy of publication after some minor aspects have been addressed.

1. This study relies on a set of observations that is unique. This is probably the reason why it is applied on a single event. It is recommended to discuss the representativeness of the results, i.e. to what extend the Z-S and DWR-S relations developed in this study can be applied to other precipitation events, regions or environmental conditions.

2. What are the perspectives in terms of implementing such approach to other instruments on the ground or in space (i.e. GPM dual-frequency radar)?

3. P.2 ll.10-11: "it is shown that a physically consistent representation of the geometric, microphysical, and scattering properties needed for radar-based QPE can be achieved" and following discussion on Ze-SR relations. For information this has been also been shown in a recent contribution involving dual-polarization ground-based radars: Bukovčić, P., A. Ryzhkov, D. Zrnić, and G. Zhang, 2018: Polarimetric Radar Relations for Quantification of Snow Based on Disdrometer Data. J. Appl. Meteor. Climatol., 57,103–120, https://doi.org/10.1175/JAMC-D-17-0090.1

4. P.3 ll.8&10: Dm is not measured; it is actually estimated from measurements.

5. p.3 ll.1-11: this paragraph seems too technical in the introduction section. You can consider including it in the methodology section.

6. Please correct Skolfronik-Jackson et al. (2015) to Skofronick-Jackson et al. (2015) throughout the paper.

7. p.13 ll.20 – p.14 l.10:  c "... Fig. 13 that there is considerable scatter at Ku-band for all three methods with the normalized standard deviation (NSTD) ranging from 55-70%". Are the errors in table 3 assumed to be normally distributed? Kirstetter et al. (2015) proposed a probabilistic Z-S QPE approach showing that uncertainty is characterized by non-symmetric distributions: Kirstetter, P.E., J.J. Gourley, Y. Hong, J. Zhang, S. Moazamigoodarzi, C. Langston, A. Arthur, 2015: Probabilistic Precipitation Rate

Estimates with Ground-based Radar Networks. Water Resources Research, 51, 1422-1442. doi:10.1002/2014WR015672 • "To reduce error, we may take the geometric mean of these two estimators": do you mean to reduce the bias? • Does the non-linear least square fitting approaches assumes normally distributed uncertainty? Can this assumption be discussed?

8. p.14 l.4: "the SR (ZKu ,DWR ) using LM method has the smallest NSTD (28.49%) but the other two methods have similar values of NSTD ($\approx$30%)". Is this difference in NSTD significant?

———————————————————

---

## Author Response (AR1)

Title: **Dual-Wavelength Radar Technique Development for Snow Rate Estimation: A Case Study from GCPEx**
Author(s): **Gwo-Jong Huang et al.**
MS No.: **amt-2018-211**
MS Type: **Research article**
Iteration: **Revised Submission**

**Authors' Response to Interactive Comments from Anonymous Referee #1 and Anonymous Referee #2**

This document combines previously posted authors' responses to comments from both Referees, as well as a marked-up revised manuscript version.

**Authors' Response to Interactive Comment from Anonymous Referee #1**

This article describes a dual-frequency method to estimate the amount of snowfall from radar measurements obtained by NASA's D3R radar during the GCPEX experiment in 2012. The proposed method hinges on a representation of the Z-SR relationship conditional to the DWR. The Authors demonstrate the superiority of their DWR-based algorithm when compared to traditional power-law relationships to retrieve the liquid-equivalent snow rate.

The article provides a nice illustration of the use of in situ microphysical data, with radiative-transfer models (T matrix with various assumptions about the mass-size relationship) constrained by remote sensing observations. However, before I can recommend this article for publication, the Authors should revise a few key points detailed below. The writing is of unequal quality with some paragraph extremely well written when others a filled with typos and unclear sentences. It would also be possible good to reduce the length of the manuscript by removing 1 or 2 figures and the overly long part that details the processing of the 2DVD images. Lastly, I couldn't find much about the efforts of the Authors to avoid or mitigate the effects of attenuation on the radar measurements, particularly at Ka band. Failing to do so can significantly bias the retrievals performed using the radar observations.

Our Response:
We would like to thank the reviewer for the positive general comments and detailed technical and editorial comments (corrections, suggestions, and questions). We have responded below to all of the comments (except those that are entirely complimentary) and included changes in the revised manuscript. In addition, we have tried to make the quality of writing more uniform in the revised manuscript.

Detailed comments and suggestions (technical questions in italic)

- General comment
  - o Please decide whether to use "gauge" (preferred) or "gage" and use this consistently throughout the manuscript. Similarly for mis-/miss-/mismatch
    Response:
    We have replaced "gage" with "gauge" and "miss-" with "mis-" on all occasions in the revised manuscript.
  - o Please provide a table of acronyms and symbols.
    We have included a table of acronyms and symbols in the revised manuscript, as *Appendix: List of Acronyms and Symbols.* The table might not be exhaustive, but it should increase the readability for the readers who are not fully familiar with GCPEx, 2DVD, and DWR retrievals.
    - Introduction o Page 2 Line12 (P2L12): "In this study, we" …
      - o P2L29: Please introduce the "Dual-\lambda" notation (e.g. in line 26) before using it.
        Response:
        We have replaced "The dual-wavelength reflectivity ratio (DWR) radar-based QPE was proposed by …" by "The dual-wavelength reflectivity ratio (DWR; the ratio of reflectivity from two different bands) radar-based QPE was proposed by …" in the revised manuscript.
      - o P3L4: "ratio of 4th moment to 3rd moment… ": "Dm" the mean mass-weighted diameter will be the ratio of the 4th to the 3rd moment only if the exponent of the m-D relationship is 3, which is seldom the case when dealing with snow… please reformulate.
        Response:
        The PSD we discussed here is in term of liquid-equivalent size (or melting size).  So the $3^{rd}$ moment of PSD is proportional to mass.
      - o P3L14: "estimate the mass of"
        Response:
        Corrected.
      - o P3L16: "based on a particle's mass"
        Response:
        Changed to "based on particles' mass".

    - Section 2

      - o P3L29: "dominates"
        Response:
        Corrected.
      - o P4L16: please include a space between "\delta_0" and "in Bohn";
        Response:
        Corrected.

- P4L19: "parameterization error": Is this an error (i.e. producing wrong results) or a different convention?
  Response:
  It is an error due to use of an equation to represent real data.
- P5L15: "circumscribeding circle"?
  Response:
  Replaced by "circumscribed circle" in the revised manuscript.
- P5, last paragraph to the end of Section 2.2: This paragraph provides waaay too much detail on the operation of the 2DVD. Please refer to literature or move to an appendix. Similarly, I'm not sure that Figure 2 is really needed…
  We believe that this paragraph and Figure 2 are really important for the paper and the explanation of geometric and fall speed measurements as a key component of the proposed and presented methodology. We have therefore opted to keep them in the revised manuscript.
- P5L21&28: "optical planes"?
  Response:
  Yes. The optical plane refers to the plane which optical system (light source and line-scan camera) is observing.
- P5L31 to 34: is it miss-match or mismatch?
  Response:
  We have corrected "miss-" to "mis-".
- P6L2: "manufacturer's matching"
  Response:
  Corrected.
- P6L8-10: "The Huang… because… size". I do not understand the causality in this sentence. Please rephrase.
  Response:
  This has been rephrased to "The Huang and Bringi approach (Huang et al. 2015) is referred to as HB …" in the revised manuscript.
- P6L26: "particles, which … Hence, the"
  Response:
  Corrected.
- P6L4: "Here, we use"
  Response:
  Corrected.
- P6L22: There is a typo in the units reported (200 and 400 \mu m ?)
  Response:
  The units are "microns".  We have corrected this typo.
- P6L24: "spheroids"
  Response:
  Corrected.

- ○ Fig3: Since both quantities being plotted are positive, would a plot in log-log scale be more appropriate/revealing?
  Response:
  The fall speed versus size plots are commonly presented in linear scale.

- Section 3

  - ○ P8L8-9: Is Figure 4 necessary? It is already in Skofronick et al 2015… Please consider deleting this Figure otherwise.
    Response:
    We think this figure is necessary to orient the reader to GCPEx and the spatial configuration of the instrumentation sites and important natural features.
  - ○ P8L10: "with an extensive"
    Response:
    Corrected.
  - ○ P8L14: "similar to a 'vertical pointing"
    Response:
    Corrected.
  - ○ P8L20: "Echo tops… high-altitude radar was were"
    Response:
    Corrected.
  - ○ Compared to Section 3.1, Section 3.2. is really well written. The Authors provide a clear description of their thorough QC efforts of the DWR data.

  - ○ What do the Authors do to address attenuation especially at Ka band? This could have a strong effect on the DWR, right?
    Response:
    In general, the attenuation of Ku- and Ka-band cannot be ignored. The case we analyzed is a low liquid equivalent precipitation rate, dry snow event. Based on the HB method, the density-size power-law relationship for this case is $\rho = 0.19*D_{app}^{-0.8}$. This relationship is very close to Ikeda-Brandes (2007) relationship which represents typical Colorado dry aggregated snow. The density of large snowflakes (> 2 mm) is less than 0.1 g/cm3. The imaginary part of the dielectric constant for such snowflakes will be close to 0. Therefore, the attenuation can be ignored. Also we computed the k-Z relationship using three methods, and the highest attenuation for Ka-band is 0.03 dB/km at 40 dBZ and most of our data set has k < 0.015 dB/km using the LM method.
  - ○ P10L29: "amount of"
    Response:
    Corrected.
  - ○ P11L22: "particles which, in principle, could"

Response:
Corrected.

- P12L3: "Figure 11a,b compares"

Response:
Corrected.

- P12L8: "falls off as"

Response:
Corrected.

- P13L1: For the airborne radar data, please refer to APR2 and cite appropriate references.

Response:

We have added in the revised manuscript a mention of APR-2 when talking about airborne radar data, and have cited Skofronik-Jackson et al. 2015 as an appropriate reference.

- P13L2: "Skolfronik"?

Response:
Corrected.

- P13L7: "mis-match" please decide between "miss-match, mis-match and mismatch" for the entre article…

Response:

Corrected "miss-" to "mis-".

- P13L11: "found not to be not size dependent"

Response:
Corrected.

- P13L12: "factor may be size"

Response:
Corrected.

- P13L20: "It is obvious from Fig. 13 that there is considerable scatter", how about in a log-log scale, is there still a significant scatter?

Response:

Yes. The log-log plot will make it look better. However, we want to represent the error of Z-SR relationship.

- P13L21-22: ".. ranging from 55 to 70%... from 40 to 45%"

Response:
Corrected.

- P14L1: Please re-write all "Ku" or "Ka" with a capital "K" even in the subscripts

Response:
Corrected.

- P14L14: "don't do not use"

Response:
Corrected.

o P14L16-l17: There is a truly remarkable agreement between measured and simulated accumulated snowfall!

o P14L18: "Figure 15b is the same as…"
Response:
Corrected.

o P14L20: why italics in "accumulations"?
Response:
Corrected.

o P15L3: "at the same time while maintaining"
Response:
Corrected.

- Section 4

o P15L6: "is to develop a technique development for … using a scanning"
Reponse:
Corrected.

o P15L26: "a large-scale synoptic … site of CARE during GCPEx."
Response:
Corrected.

o P15L30: "explained by a possible"
Response:
Corrected.

o P16 1st paragraph: Could the discrepancies that you couldn't explain be due to attenuation, which could strongly affect the Ka-band measurements?
Response:
According to our scattering computation, the most of our data set has attenuation less than k < 0.015 dB/km. The maximum range of D3R is 30 km.  So the worst round trip PIA is 0.9 dB which is less than the error of reflectivity measurement (~1dB). Considering that the Ku-band reflectivity is also attenuated, the impact on DWR should be even less. However, we agree that for heavy snow, more compact snow types (i.e., graupel), or melting snow, an attenuation correction would be needed.

o P16L13: "and LM gave (1.94 mm)."
Response:
Corrected.

o P16L19-21: Is the smoothing of DWR actually a smoothing of Ka or is it necessary to first form DWR and then smooth only this parameter?
Response:
Since Ku- and Ka-band reflectivities are two independent measurements, it does not matter whether smoothing of Z is done first or DWR is formed first and smoothed.

o Could you also provide some perspective as to how to extend your method to other dual-frequency datasets? Would the methods presented here still work if applied to a Ku-W or Ka-W data set? What would be the expected differences/commonalities? Similarly, would there be some potential or added value for airborne or spaceborne dual-frequency radars? Lastly, how would you extend your method to handle more frequencies (3,4,..)?

Response:

Theoretically, our method is independent of the frequency band pair. However, reflectivity oscillates when the particle's size is in the Mie region, thus the DWR also oscillates. So, a different frequency band pair would have a different dynamic range of DWR.

The second problem is due to the T-matrix method. By using the HB method, in our experience, T-matrix will have convergence problem when frequency is higher than Ka-band. The LM and HW methods use Liao's model (Liao et al. 2013) for the scattering computations. They compute scattering coefficients up to 183 GHz, and their equivalent diameter limits are 2.5 mm. Therefore, this size may be too small for large snowflakes/graupel or highly rimed snowflakes.

- References

o P17L24: "for scattering"
Response:
Corrected.
o P17L31: "Orinetations"
Response:
Corrected.

- Figures

o Figure 10: legend: "that were directly measured…"
Response:
Corrected.
o Figures 13 and 14:
• Please use the same units for SR between the two plots (Figs 13 and 14) to help the reader compare plots;
Response:
This was a mistake. The unit of SR in the figures should be mm/hr. We have corrected Figure 13.
• In Fig. 13, would a logarithmic scale work better?
Response:
We believe that it is better to use linear scale.

**Authors' Response to Interactive Comment from Anonymous Referee # 2**

A method is proposed to estimate snowfall rates from the D3R dual-frequency radar measurements. The method is developed and assessed during a snowfall event in 2012. 2D-video disdrometer and gauge data are used to link observed radar reflectivity and particles physical properties. The dual-frequency estimator is shown to estimate snowfall rates with more accuracy than the conventional single frequency Z-S approach.

The paper is well written. The detailed discussion on the assumptions, methodology and techniques is appreciated. Dual-frequency estimation of snowfall rates addresses the critical need for improved snow estimation from ground- and space-based remote sensing. The work presented is worthy of publication after some minor aspects have been addressed.

Our Response:
We would like to thank the reviewer for the positive general comments, as well as the corrections, suggestions, and questions regarding minor aspects. We have responded below to all of the comments (except those that are entirely complimentary) and included changes in the revised manuscript.

1. This study relies on a set of observations that is unique. This is probably the reason why it is applied on a single event. It is recommended to discuss the representativeness of the results, i.e. to what extend the Z-S and DWR-S relations developed in this study can be applied to other precipitation events, regions, or environmental conditions.

Response:
The case we have analyzed during GCPEx is a common synoptic snowfall event (large scale forcing is described in brief in Section 3) that occurs in the area surrounding the CARE site and is not unique per se, but dual-wavelength scanning radar observations combined with measuring instruments sited inside a DFIR wind shield are somewhat unique. The Z-SR power law derived for this event is expected to be applicable to similar synoptic forced snowfall in similar climatology under similar environmental conditions (eg Temperature and RH profiles from sounding) but not for example to lake effect snowfall as the microphysics are quite different. Regarding the DWR-SR relations, they should be considered for now as 'condition' specific and analysis of more events are needed before any firm conclusions can be drawn as to applicability to other regions or environmental conditions.

We have added some sentences in the conclusions at the very end as the ending para.

*"The snow rate estimation algorithms developed here are expected to be applicable to similar synoptic forced snowfall under similar environmental conditions (e.g., temperature and relative humidity) but not for example to lake effect snowfall as the microphysics are quite different. However, analysis of more events are needed before any firm conclusions can be drawn as to applicability to other regions or environmental conditions."*

2. What are the perspectives in terms of implementing such approach to other instruments on the ground or in space (i.e. GPM dual-frequency radar)?

Response:
Substantial work exists and is on-going using airborne particle size distribution probes and downward pointing radars to develop Z-SR relations stratified by temperature, for example. Multiple-wavelength downward pointing airborne radars at the GPM frequencies (Ku, Ka-bands) have been used in many field programs including comparisons with GPM overpasses. The GPM dual-wavelength technique for measuring snowfall near the surface is an active area of research. As such our experience using scanning dual-wavelength radar (D3R) is bound to be useful as ground-validation for GPM-based algorithms.

3. P.2 ll.10-11: "it is shown that a physically consistent representation of the geometric, microphysical, and scattering properties needed for radar-based QPE can be achieved" and following discussion on Ze-SR relations. For information this has been also been shown in a recent contribution involving dual-polarization ground-based radars: Bukovci ̆ c, P., A. Ryzhkov, D. Zrni ́ c, and G. Zhang, 2018: Polarimetric Radar ́Relations for Quantification of Snow Based on Disdrometer Data. J. Appl. Meteor. Climatol., 57,103–120, https://doi.org/10.1175/JAMC-D-17-0090.1.

Response:
We are aware of this paper which uses dual-polarization radar to estimate snowfall with algorithms developed using 2DVD. To the best of our knowledge, the derived algorithms were not tested with independent snow gage measurements. We have cited this reference in our revised version on page 2, lines 10-11.

4. P.3 ll.8&10: Dm is not measured; it is actually estimated from measurements.

Response:
We have changed "measurements" to "estimation".

5. p.3 ll.1-11: this paragraph seems too technical in the introduction section. You can consider including it in the methodology section.

Response:
After much consideration we feel that keeping the paragraph the same i.e., in the Introduction, is reasonable.

6. Please correct Skolfronik-Jackson et al. (2015) to Skofronick-Jackson et al. (2015) throughout the paper.

Response:
This typo has been corrected throughout the paper.

7. p.13 ll.20 – p.14 l.10: Fig. 13 that there is considerable scatter at Ku-band for all three methods with the normalized standard deviation (NSTD) ranging from 55- 70%". Are the errors in table 3 assumed to be normally distributed? Kirstetter et al. (2015) proposed a probabilistic Z-S QPE approach showing that uncertainty is characterized by nonsymmetric distributions: Kirstetter, P.E., J.J. Gourley, Y. Hong, J. Zhang, S. Moazamigoodarzi, C. Langston, A. Arthur, 2015: Probabilistic Precipitation Rate C2 Estimates with Ground-based Radar Networks. Water Resources Research, 51, 1422-1442. doi:10.1002/2014WR015672 "To reduce error, we may take the geometric ´ mean of these two estimators": do you mean to reduce the bias? Does the non- ´ linear least square fitting approaches assumes normally distributed uncertainty? Can this assumption be discussed?

*Response:*
As stated in the text the Z-SR power law prefactor and exponent are determined using weighted total least squares method where Z is in $mm^6m^{-3}$ and SR in mm/h. We do not fit a straight line in log-log space. As in all least squares methods the residual errors are assumed to be normally distributed. Also, the estimated NSTD values reflect the parameterization errors only (see Chapter 7 of Bringi and Chandrasekar 2001) in an additive error model. We use the geometric mean to show the central tendency and avoid giving too much weight to the outliers. It is not intended for reducing the bias. We do not believe that it is necessary to include this in the revised text.

We have read the article by Kirstetter et al. which is based on mapping the radar measurement of Z to measured snow rate at the ground; thus their uncertainty is due to various error sources such as radar bias in Z as well as errors in snow gage measurements due to horizontal wind etc. They find 50% bias (underestimate) when using the climatological Z-SR relationship and the relative uncertainty expressed in terms of the interquartile range normalized by the mean of 50-75%. We agree that the errors (bias+random) can be unsymmetric and interquartile ranges are more appropriate. We have added a few sentences from Kirstetter et al. in our revised manuscript just before Section 4 on Summary and Conclusions.

*"Note that the error model used here is additive with the parameterization and measurement errors modeled as zero mean and uncorrelated with the corresponding error variances estimated either from data or via simulations (as described in Chapter 7 of Bringi and Chandrasekar 2001). This is a simplified error model since it assumes that radar Z and snow gage measurements are unbiased based on accurate calibration. A more elaborate approach of quantifying uncertainty in precipitation rates is described by Kirstetter et al. (2015)."*

8. p.14 l.4: "the SR (ZKu ,DWR ) using LM method has the smallest NSTD (28.49%) but the other two methods have similar values of NSTD (≈30%)". Is this difference in NSTD significant?

*Response:*
The differences in NSTD are not statistically different. That sentence has been modified to:

[revised manuscript text omitted]

---

## Author Response (AR2)

Title: **Dual-Wavelength Radar Technique Development for Snow Rate Estimation: A Case Study from GCPEx**
Author(s): **Gwo-Jong Huang et al.**
MS No.: **amt-2018-211**
MS Type: **Research article**
Iteration: **Revised Submission – Revision 2, January 12, 2019**

**Authors' Response to Comments from Anonymous Referees**

This document combines authors' responses to comments from Associate Editor and both Referees, as well as a marked-up revised manuscript version (Revision 2).

**Authors' Response to Comments from Associate Editor**

Comments to the Author:
Please carefully consider all of the points raised by Anonymous Referee #3 (Report #2) and address these in response and a revised manuscript. A final editorial decision will be reached after these items are received.

> Our Response:
>
> We have responded to all comments by the reviewers and have revised the manuscript accordingly. All revisions are highlighted yellow in the revised manuscript (Revision 2), and are explained in this document.

**Authors' Response to Comments from Anonymous Referee #2 (Report #1)**

*No new comments.*

> Our Response:
>
> We are glad that the Referee is happy with our Revised Manuscript (Revision 1).

**Authors' Response to Comments from Anonymous Referee #3 (Report #2)**

This manuscript describes a methodology for snow rate estimation from dual frequency (Ku- and Ka-band) ground radar observations. Overall, I find the manuscript informative and well written. Nevertheless, the following minor issues need to be considered before publication:

> Our Response:
>
> We have responded below to the comments (minor issues) and included changes in the revised manuscript (Revision 2).

1) It is not apparent from Figure 3 and the associated text (page 6, Second paragraph) what is the fraction of particles excluded from the analysis. This is particularly important for the understanding of Fig. 10.

Response:

The green circles in Figure 3 show the fall speed versus size ($D_{app}$) based on the manufacturer's matching code (used only for methods 2 and 3 as described in Section 3.3). This code found 507,833 matched particle. Many studies have shown that the manufacturer's matching code has significant mis-matching problem which will cause over-estimation of fall speed. Correct fall speed is the key factor to compute particles' mass. Therefore, we used Hanesch (1999) scan line criteria to filter out possible mis-matched particles. We found that there are 175,199 (34.5%) particles which satisfy the criteria. These particles are shown as "magenta +" in Figure 3.  We compute the mass of these 175,199 particles by using Böhm equations (LM method) and divide mass by apparent volume ($\pi*D_{app}^3/6$) to get the effective density. The mass computed by the Heymsfield-Westbrook equations are very similar to Böhm. Since the density of ice particles cannot exceed 0.92 g cm$^{-3}$, we delete those particles whose densities are larger than 1 g cm$^{-3}$. The remaining particles are 128,063 (25.2%) which are the blue "x"s in Figure 3. The results shown in this paper which use methods 2 and 3 in Section 3.3 are based on these 128,063 particles.

We have clarified the text in Section 2.2 in the revised manuscript (Revision 2) as follows (highlighted yellow below) to explain the differences in the PSD adjustment factor for method 1 versus methods 2 and 3 (described in Section 3.3):

"In the Appendix of Huang et al. (2010), they showed that mis-match will cause the volume, vertical dimension, and fall speed of particles to be over-estimated.  Subsequently, the mass of particles will also be over-estimated mainly because of fall speed.  To get the best estimation of mass, they used 2DVD single camera data and re-did the matching based on a weighted Hanesch criteria (Hanesch 1999). If the match criteria are not satisfied, then that particle is rejected; it follows that the concentration will tend to be underestimated. To readjust the measured concentration for this underestimate (assumed to be a constant factor), the procedure described in Huang et al. (2015) is used which only involves the ratio of the total number of particles counted in the scan area of the single camera to the number of successfully matched particles in the virtual measurement area. For the event analysed here (using method 1 in Section 3.3), this adjustment factor is between 1.1 and 1.5. The Pluvio gauge accumulation is not used as a constraint in method 1."

Further, in Section 2.2 in the revised manuscript (Revision 2) we have clarified the use of the manufacturer's matching code as follows (highlighted yellow below):

"For methods 2 and 3 in Section 3.3 we used the manufacturer's matching algorithm which gives the contour data……..   To obtain reliable fall speeds, we examined all matched particles (given by the manufacturer's matching algorithm numbering 507,833 for the event and marked as green in Fig. 3) and removed those particles which did not satisfy the Hanesch scan line criteria resulting in 175,199 (34.5%) of particles that did satisfy the match criteria (magenta marks in Fig. 3). We used the fall speed of these filtered particles to compute their mass for both Böhm and Heymsfield-Westbrook methods, and then, dividing the mass by apparent volume ($= \square D_{app}^3/6$), to get the particle density.  Since the maximum density of ice particles is around 0.9 g cm$^{-3}$, we further remove particles whose density comes out as larger than 1 g cm$^{-3}$.  After this two-step

filtering, the particles we use for further analysis (numbering 128,063) are shown in Fig. 3 as blue points. The filtering will eliminate particles, which will reduce the liquid equivalent snow accumulation. Hence, the Pluvio gauge accumulation is used as an integral constraint, i.e., the concentration in each bin is increased by a constant factor to match the 2DVD accumulation to the Pluvio accumulation. This constraint is only used in methods 2 and 3 in Section 3.3."

Again in Section 3.3 we have clarified that the PSD adjustment factor determination for method 1 is different than for methods 2 and 3 [please see the multiple added pieces of text highlighted yellow in Section 3.3 in the revised manuscript (Revision 2)].

Finally, we have revised the caption to Figure 3 which now reads:

**Figure 1:** Fall speed versus $D_{app}$ for the synoptic case on January 31, 2012 at the CARE site. The green circles represent the results of the manufacturer's matching algorithm which is known to allow mis-matched particles with unrealistic fall speeds. The first filtering step is the selection of matched particles which satisfy Hanesch scan line criteria (magenta). The second filter step is shown as blue "x"'s which are based on particles whose density (from mass computed by Böhm's or Heymsfield-Westbrook method) is lower than 1 g cm$^{-3}$.

2) While the Huang et al. (2015) and Lia et al. (2013) scattering models are reasonable and computationally simple, so is the the Rayleigh-Gans based approach of Westbrook et al. (2006;2008). This approach is based on the fact that snow particles are characterized by a relative refractive index close to 1.0, which makes it possible to approximate their radar cross-sections as the product of a constant, their squared mass and a form factor that depends on the parameter size. Given the fact that, physically, the Rayleigh-Gans approximation provides more insight into the electromagnetic properties of snowflakes than the empirical evidence that a certain aspect ratio or a certain density may work, the authors should at least compare the radar cross-sections derived from Westbrook et al. (2008) parameterization to those from Huang et al. (2015) and Liao et al. (2013), if not include the fourth approach in Section 3.3.

Westbrook, C. D., R. C. Ball, and P. R. Field, 2006: Radar scattering by aggregate snowflakes. Quart. J. Roy. Meteor. Soc., 132, 897–914.
Westbrook, C. D., R. C. Ball, and P. R. Field, 2008: Corrigendum: Radar scattering by aggregate snowflakes. Quart. J. Roy. Meteor. Soc., 134, 547–548.

We agree that the Westbrook et al. approach gives good physical insight into the scattering process, and the two papers are now cited in Section 2.3 in the revised manuscript (Revision 2) as:

"Westbrook et al. (2006; 2008) used the Rayleigh-Gans approximation to develop an analytical equation for the scattering cross sections of simulated snow aggregates of bullet rosettes using an empirical fit to the form factor that accounts for deviations from the Rayleigh limit."

Their approach is based on a number of assumptions based on simulated shapes of unrimed aggregates of bullet rosettes following a mass-radius of gyration (r) power law where the exponent is 2. We cannot compute the r from 2DVD contour data and cannot confirm that r=0.3*D. The form factor for non-Rayleigh sizes in their eq (16) is 'tuned' for their simulated aggregates with coefficients $c_1$ and $c_2$.

Nevertheless, we used their eq (16) to compute DWR and compared with the radar measurements (see Figure R1 below). The best agreement was obtained using $c_1$=12.7 and $c_2$=3.6 (and not $c_2$=4.5 as given in the Corrigendum). The calculated DWR from Westbrook et al. eq (16) is biased too high relative to the radar measurements as well as the three methods especially from 0100-0300 UTC. The reason for this significant bias in computed DWR is not clear but might be related to the fact that our exponent of the mass-dimensional relation based on fall speeds was around 2.5 indicative of some riming and observers at the CARE site reported aggregates of dendrites as opposed to bullet rosettes.

The new DWR calculations were done only as a response to the reviewer's comments. We do not believe that it is necessary to include it in our revised manuscript as it is based on assumptions that are likely not applicable to our dataset. Our goal was to get the PSD and mass-dimensional relations inferred from 2DVD measurements of fall speed, geometry, etc., and not relying on simulated aggregate models. As future work we hope to use scattering lookup tables based on DDA covering a wider spectrum of particle shapes and degree of riming.

[Figure]

Figure R1: Time series of DWR using the new formula of Westbrook et al. (2008) compared with radar measurements and the three other methods based on 2DVD simulations.

[revised manuscript text omitted]